# Enhancing Inverse Modeling in Groundwater Systems through Machine Learning: A Comprehensive Comparative Study

Junjun Chen[1,2], Zhenxue Dai[2,3], Shangxian Yin[4], Mingkun Zhang[5], Mohamad Reza Soltanian[6]

[1] National and Local Joint Engineering Laboratory of Internet Application Technology on Mine, China University of Mining and Technology, Xuzhou, 221008, China

[2] College of Construction Engineering, Jilin University, Changchun, 130026, China

[3] School of Environmental and Municipal Engineering, Qingdao University of Technology, Qingdao, 273400, China

[4] College of Safety Engineering, North China Institute of Science and Technology, Langfang, 065201, China

[5] Shandong Rui Yi technology development Co., Ltd., Jinan, 250000, China

[6] Departments of Geosciences and Environmental Engineering, University of Cincinnati, OH, 45220, USA

*Correspondence to*: Zhenxue Dai (dzx@jlu.edu.cn), Shangxian Yin (yinshx03@126.com)

**Abstract.** Tandem neural network architecture (TNNA) is a machine learning algorithm which has been recently proposed for estimating uncertain parameters with inverse mappings. However, its reliability has only been validated in limited research scenarios, and its advantages over conventional methods remain underexplored. This study systematically compares the performance of the TNNA algorithm with four traditional metaheuristic algorithms across three heterogeneity scenarios, each employing a specific inversion framework: (i) a surrogate model coupled with an optimization algorithm for cases with eight homogeneous parameter zones, (ii) Karhunen-Loève Expansion (KLE)-based dimensionality reduction combined with a surrogate model and an optimization algorithm for a high-dimensional Gaussian random field, and (iii) generative machine learning-based dimensionality reduction integrated with a surrogate model and an optimization algorithm for a high-dimensional non-Gaussian random field. Additionally, we evaluate algorithm performance under two different noise level conditions (multiplicative Gaussian noise with standard deviations of 1% and 10%) for normalized hydraulic head and solute concentration data in the non-Gaussian random field scenario, which exhibits the most complex parameter characteristics. The results demonstrate that both the TNNA algorithm and the metaheuristic algorithms achieve inversion results that satisfy the convergence accuracy within these machine learning- based inversion frameworks. Moreover, under the 10% high-noise condition in the non-Gaussian random field, the inversion results remain robust when sufficient constraints are imposed. Compared to metaheuristic approaches, the TNNA method yields more reliable inversion results with significantly higher computational efficiency, highlighting the considerable advantages of machine learning in advancing groundwater system inversions.

## 1 Introduction

Numerical models are essential for quantifying flow and mass transport dynamics within aquifers, providing significant insights into hydrological and biogeochemical processes (Steefel et al., 2005; Sanchez-Vila et al., 2010; Sternagel et al., 2021; Xu et al., 2022). However, directly measuring aquifer parameters, such as permeability fields, remains challenging due to limitations in current hydrogeological exploration techniques and budgetary constraints (Yeh, 1986; Kool et al., 1987; Beven and Binley, 1992; Mclaughlin and Townley, 1996; Dai and Samper, 2004; Castaings et al., 2009; Chen et al., 2021). Inverse modeling has become a key approach for estimating these uncertain model parameters, improving the accuracy of numerical simulations (Ginn and Cushman, 1990; Carrera and Glorioso, 1991; Hopmans et al., 2002; Zheng and Samper, 2004; Zhou et al., 2014; Bandai and Ghezzehei, 2022; Abbas et al., 2024; Giudici, 2024).

Inverse modeling within Bayesian theorem-based data assimilation frameworks has garnered significant attention from the hydrogeological community over the past few decades (Scharnagl et al., 2011; Chen et al., 2013; Zhang et al., 2018; Xia et al., 2021). Among available algorithms, methods based on objective functions established from maximum a posteriori estimation and solved by optimization techniques represent a significant category (Tsai et al., 2003; Blasone et al., 2007; Sun, 2013; Vrugt, 2016). One type is local optimization algorithms, which update model parameters from initial guesses towards optimal solutions according to gradient directions, such as the Gaussian-Newton method (Dragonetti et al., 2018; Qin et al., 2022) and the Levenberg-Marquardt method (Schneider-Zapp et al., 2010; Nhu, 2022). These methods are highly efficient but may converge to local optima when dealing with nonconvex inversion problems. Another category is to achieve global optima solutions through metaheuristic searches, which typically incorporate processes of exploration (to search the entire parameter space for a diverse range of estimates) and exploitation (to leverage local information to refine estimates). Popular metaheuristic algorithms include the Genetic Algorithm (GA) (Ines and Droogers, 2002; Lindsay et al., 2016), Simulated Annealing (SA) (Kirkpatrick et al., 1983; Jaumann and Roth, 2018), Differential Evolution (DE) (Li, 2019; Yan et al., 2023), and Particle Swarm Optimization (PSO) (Rafiei et al., 2022; Travaš et al., 2023). Nevertheless, their computational efficiency may be reduced by extensive exploration and exploitation processes in achieving globally optimal inversion results. Accurate and efficient estimation of uncertain model parameters across various scenarios remains one of the most significant challenges for developing inversion frameworks.

In recent years, machine learning has experienced rapid developments and demonstrated significant performance in addressing complex problems characterized by high dimensionality and nonlinearity (Hinton and Salakhutdinov, 2006; Lecun et al., 2015; Bentivoglio et al., 2022; Shen et al., 2023). Integrating conventional inversion methods with cutting-edge machine learning techniques has become increasingly popular in addressing the challenges of inversion studies. One effective strategy is constructing surrogate models to accelerate forward simulations, ensuring that inversion algorithms perform comprehensive searches across the entire parameter space more efficiently (Razavi et al., 2012). For instance, Zhan et al. (2021) identified lithofacies structures by utilizing a deep octave convolution residual network to construct a surrogate model for predicting solute concentrations and hydraulic heads in heterogeneous aquifers. Wang et al. (2021) constructed a subsurface flow

surrogate model under heterogeneous conditions through physically informed neural network methods, specifically for uncertainty quantification and parameter inversion. Liu et al. (2023) constructed a convolutional neural network (CNN) surrogate model to combine with a hierarchical homogenization method to estimate effective permeability of digital rocks. More related studies can also be found in recent reviews (Yu and Ma, 2021; Luo et al., 2023; Zhan et al., 2023). Specifically, inversion approaches based on objective function minimization can also benefit from adjoint methods (Plessix, 2006). Integrating adjoint methods with machine learning-based surrogate models enables efficient gradient computation in high-dimensional and complex scenarios, making their practical implementation tractable (Xiao et al., 2021).

In addition to surrogate models, parameter optimization through machine learning-based reverse mapping represents another significant advancement in inversion techniques. Previous studies have outlined at least two strategies to achieve reverse mapping models. The first strategy is the data-driven approach, where reverse regressions are trained using datasets that comprise pairs of model outputs and inputs. For example, Sun (2018) developed a regression model from hydraulic heads to heterogeneous conductivity fields using a CNN-based generative adversarial network (GAN) approach. Kuang et al. (2021) succeeded in real-time identification of earthquake focal mechanisms by training a deep neural network (DNN) regression on seismic waveform data. Yang et al. (2022) established the relationship between gravity data and $CO_2$ plumes to perform real-time inversion for geologic carbon sequestration. Another strategy is to train a reverse network within the tandem neural network architecture (TNNA) integrated with a pre-trained surrogate model (i.e., forward network). The TNNA method was introduced with the advent of deep learning and has been successfully applied in computed tomography reconstruction (Adler and Öktem, 2017), nanophotonic structure inverse design (Liu et al., 2018; Yeung et al., 2021), and photonic topological state inverse design (Long et al., 2019). Our previous research expanded the application of the TNNA algorithm within groundwater science, evaluating its performance in reactive transport inverse modeling and improving inversion results by incorporating an adaptive update strategy to reduce local predictive errors of surrogate models. The findings indicated that accurate surrogate model predictive results around the actual parameter values yield dependable TNNA inversion outcomes (Chen et al., 2021).

The TNNA algorithm demonstrates a fundamental advantage by requiring only a single forward simulation to update parameters in each iteration. In contrast, conventional metaheuristic algorithms typically necessitate multiple forward simulations. Despite the innovation of this approach, its applicability in more general groundwater numerical scenarios and its performance compared to conventional metaheuristic algorithms remain uncertain. This study considers three cases with different heterogeneity characteristics to compare the performance of the TNNA algorithm with four conventional metaheuristic algorithms. In Case 1, the domain is divided into a finite number of homogeneous zones. The other two Cases focus on high-dimensional parameter fields based on the spatial variability of the aquifer. These two cases are essential for revealing the dynamic behaviors of the groundwater system at the discrete grid scale. Depending on the spatial variability of the aquifer structure, the two high-dimensional numerical cases characterize the heterogeneity of aquifer parameters using a Gaussian random field (i.e., Case 2) and a non-Gaussian random field (i.e., Case 3), respectively. The Gaussian random field is suited for aquifers with a single lithofacies and relatively uniform physical structures, where the spatial variation of parameter values is quite smooth. In contrast, the non-Gaussian random field accounts for the existence of a nugget effect in the aquifer

structure, such as when it contains multiple lithofacies with varying hydraulic properties (Mariethoz and Caers, 2014). For comparative study of the three Cases, surrogate models will be used to accelerate forward simulation. Additionally, dimensionality reduction techniques are necessary for the two high-dimensional cases to reduce computational complexity associated with high-dimensional parameter spaces. Specifically, the Karhunen-Loève Expansion (KLE) method is feasible for Gaussian random fields. It reconstructs the Gaussian random field through a linear combination of orthogonal basis functions, achieving dimensionality reduction by retaining the dominant modes corresponding to the largest eigenvalues (Loève, 1955; Zhang and Lu, 2004; Mariethoz and Caers, 2014). However, the second-order statistics relied upon by KLE are insufficient to fully represent complex characteristics for non-Gaussian random fields. In recent years, generative machine learning methods have demonstrated outstanding performance in parameter field reconstruction (Mo et al., 2020; Zhan et al., 2021; Guo et al., 2023). These methods can establish relationships between low-dimensional standard distributions (e.g., uniform distribution) and high-dimensional distributions, effectively representing non-Gaussian random fields as low-dimensional latent vectors (i.e., parameters after dimensionality reduction). Thus, extending the TNNA framework by integrating KLE and generative machine learning methods, respectively, is a potentially feasible approach for solving the high-dimensional heterogeneous aquifer parameter inversion problems presented in Case 2 and Case 3. In summary, the primary contributions of this study are as follows:

(1) Proposed a novel inversion framework that integrates the TNNA algorithm with dimensionality reduction techniques, including KLE and generative machine learning methods, thereby extending its applicability to high-dimensional heterogeneous fields characterized by Gaussian and non-Gaussian stochastic processes, respectively.

(2) Conducted a comprehensive comparative analysis between the TNNA algorithm and four conventional metaheuristic algorithms across three case scenarios, highlighting the advantages of machine learning in inverse estimation under different heterogeneous conditions.

With advancements in artificial intelligence, the anticipated outcomes of this study are expected to significantly enhance the development of novel inversion algorithms, offering new insights for future studies. The following sections of this paper are structured as follows: Section 2 introduces the fundamental principles of the methodology involved in this study. Section 3 provides detailed information on numerical models for the three cases. Section 4 presents the results and discussions. Finally, Section 5 presents a summary and conclusions derived from this research, along with recommendations for future studies.

## 2. Methodology

The inversion framework based on nonlinear optimization theory generally consists of two aspects: (1) constructing nonlinear constraints for the optimization of uncertain model parameters, and (2) establishing optimization algorithms to search for the model parameters that satisfy these constraints. The general form of the nonlinear optimization model in this paper is as follows:

$$m^*=\min \sum_{i=1}^{N_{\text{obs}}} \frac{1}{\sigma_i}\Big[\mathbf{y}_{\text{obs}}[i] - \hat{\mathbf{y}}[i]\Big]^2$$
$$\begin{cases} \hat{\mathbf{y}}=\boldsymbol{F}_{HF}(\boldsymbol{m}) \\ \boldsymbol{m}^L \leq \boldsymbol{m} \leq \boldsymbol{m}^U \end{cases} \tag{1}$$

where $\mathbf{y}_{\text{obs}} \in \mathbb{R}^{N_{\text{obs}} \times 1}$ and $\hat{\mathbf{y}} \in \mathbb{R}^{N_{\text{obs}} \times 1}$ represent the observed data vector and the corresponding model simulation output vector, respectively. $\mathbf{y}_{\text{obs}}[i]$ and $\hat{\mathbf{y}}[i]$ refer to the $i$-th element of the observed and simulated vectors, respectively, and $\sigma_i$ denotes the standard deviation of the $i$-th observed data. $\boldsymbol{m}$ represents the vector of model parameters to be optimized, $\boldsymbol{m}^*$ denotes the optimal parameter vector obtained through optimization; $\boldsymbol{m}^L$ and $\boldsymbol{m}^U$ are the vectors representing the lower and upper limit values of the model parameters, respectively. $\boldsymbol{F}_{HF}(\cdot)$ represent the high-fidelity numerical model.

In this study, three different inversion frameworks are developed to compare the TNNA algorithms with four metaheuristic algorithms. In low-dimensional parameter scenario, a surrogate model $\boldsymbol{F}_{Forward}(\cdot)$ is constructed to approximate high-fidelity numerical prediction outputs. Therefore, the objective function of the inversion framework integrated with a surrogate model is as follows:

$$m^*=\min \sum_{i=1}^{N_{\text{obs}}} \frac{1}{\sigma_i}\Big[\mathbf{y}_{\text{obs}}[i] - \boldsymbol{F}_{Forward}(\boldsymbol{m})[i]\Big]^2 \tag{2}$$

In high-dimensional parameter scenarios, in addition to employing surrogate models, dimensionality reduction algorithms are also integrated in inversion frameworks. Let $\boldsymbol{m}=\boldsymbol{G}(\boldsymbol{z})$ represent an operator for parameter dimensionality reduction. Specifically, the Karhunen-Loève Expansion (KLE) and the Octave Convolution Adversarial Autoencoder (OCAAE) are used for representing Gaussian random fields and non-Gaussian random fields, respectively. The high-dimensional parameter $\boldsymbol{m}$ is optimized indirectly by estimating the low-dimensional vector $\boldsymbol{z}$:

$$z^*=\min \sum_{i=1}^{N_{\text{obs}}} \frac{1}{\sigma_i}\Big[\mathbf{y}_{\text{obs}}[i] - \boldsymbol{F}_{Forward}(\boldsymbol{G}(\boldsymbol{z}))[i]\Big]^2$$
$$\boldsymbol{m}^* = \boldsymbol{G}(\boldsymbol{z}^*) \tag{3}$$

The basic mathematical theories of surrogate models, dimensionality reduction techniques, and optimization algorithms are introduced in Section 2.1 to 2.3, respectively.

## 2.1 Surrogate modeling methods

In this study, surrogate models $\boldsymbol{F}_{Forward}(\cdot)$ are developed using a data-driven strategy as shown in Figure 1. The process begins by sampling model parameters from prior distributions and calculating their responses using high-fidelity numerical models. A training dataset of paired model parameters and responses is then obtained, which is used to construct surrogate models via supervised machine learning. Specifically, four popular machine learning models with distinct architectural differences are evaluated for surrogate modeling. These are: multi-output support vector regression (MSVR), a kernel-based architecture for data mapping; fully connected deep neural network (FC-DNN), composed of stacked fully connected layers;

LeNet, a classical convolutional neural network (CNN) architecture; and deep residual convolutional neural network (ResNet), which incorporates residual connections into the CNN structure.

The detailed principles of MSVR and the three deep learning-based methods are illustrated in the following two subsections. The predictive accuracy of four surrogate modeling approaches will be compared in this study, and the best-performing approach among them will subsequently be selected for inversion computations. Before constructing surrogate
models, the training datasets are normalized separately for each simulation component using Min-Max Normalization, in which each component is scaled independently based on its minimum and maximum values, ensuring that all normalized values fall within the range [0, 1] (Chen et al., 2021).

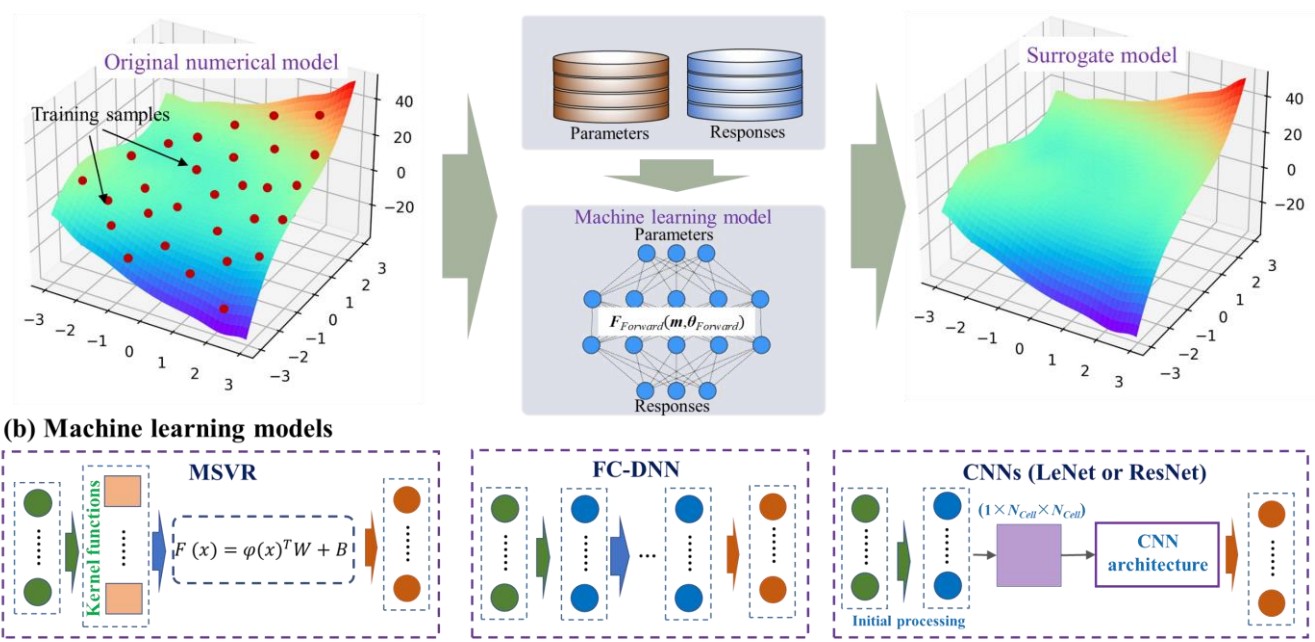

**Figure 1**. The framework for data-driven based surrogate model construction and the machine learning models employed.
Note that for CNN-based surrogate models, the initial processing module is activated only for low-dimensional scenarios, whereas in high-dimensional scenarios, the parameter matrix ($1 \times N_{Cell} \times N_{Cell}$) is directly input into the CNN architecture.

### 2.1.1 MSVR

MSVR is developed from the original support vector machine (SVM) for realizing multivariate regression (Pérez-Cruz et al., 2002; Tuia et al., 2011). The mathematical expression is given as follows:

$$\hat{\mathbf{y}} = F_{MSVR}(\mathbf{m}) = \varphi(\mathbf{m})^T W + B \tag{4}$$

where $F_{MSVR}(\cdot)$ denotes the dataset regression model operator constructed based on MSVR; $\varphi(\mathbf{m})$ is a nonlinear regression function that implicitly maps the input vector $\mathbf{m}$ into a high-dimensional feature space. Its inner product defines the kernel function $K(\mathbf{m}, \mathbf{m}_i)$ (here, we use the Gaussian radial basis function (RBF) kernel with a bandwidth parameter $\sigma$: $K(\mathbf{m}, \mathbf{m}_i) =$

$\varphi(\boldsymbol{m})^T\varphi(\boldsymbol{m}_i) = \exp(-0.5 \parallel \boldsymbol{m} - \boldsymbol{m}_i \parallel^2/\sigma^2))$. Assuming $N_{samples}$ denotes the number of surrogate model training samples, the regression coefficients $W=[w^1,\ldots,w^{N_{obs}}]^T \in \mathbb{R}^{N_{obs} \times N_{samples}}$ and $B=[b^1,\ldots,b^{N_{obs}}]^T \in \mathbb{R}^{N_{obs} \times 1}$ are determined by minimizing the structural risk, as outlined in equations (5) and (6):

$$W,B = argminL(W,B) = \frac{1}{2}\sum_{j=1}^{N_{obs}} \left\| w^j \right\|^2 + C \sum_{j=1}^{N_{samples}} L(u_i) \tag{5}$$

where $C$ is a penalty parameter; and $L(u)$ is a quadratic $\varepsilon$-insensitive loss function, expressed as:

$$L(u) = \begin{cases} 0, & u < \varepsilon \\ (u-\varepsilon)^2, & u \geq \varepsilon \end{cases} \tag{6}$$

where $u_i = \|e_i\| = \sqrt{e_i^T e_i}$; $e_i^T = y_i^T - \varphi^T(\boldsymbol{m}_i)W - B^T$. For $\varepsilon=0$, this problem is equivalent to an independent regularized kernel least square regression for each component. For $\varepsilon \neq 0$, it becomes feasible to develop individual regression functions for each dimension based on the model outputs and to generate their corresponding support vectors. Solving the optimization problem directly is challenging, and the desired solutions for $W$ and $B$ are determined using an iterative reweighted least squares (IRWLS) procedure, employing the quasi-Newton approach. During the IRWLS process, the term $L(u)$ in equation (5) is first transformed into a discrete first-order Taylor expansion, and the corresponding quadratic programming approximation is constructed. Meanwhile, a linear expression is derived based on the principle that the first-order derivatives of the objective function with respect to $W$ and $B$ are zero. Finally, the optimal values of $W$ and $B$ are obtained through a line search. Further details on the IRWLS procedure can be found in (Sanchez-Fernandez et al., 2004).

The performance of the MSVR model is influenced by three hyperparameters: $C$, $\sigma$ and $\varepsilon$ (Ma et al., 2022). This study optimizes these hyperparameters by minimizing the root mean square error (RMSE) using the four metaheuristic algorithms introduced in this study.

### 2.1.2 Deep learning based surrogate models

(1) DNN architectures

The three DNN models are all feedforward neural networks, which are generally constructed by stacking multiple hidden layers. The structure can be expressed as $\boldsymbol{F}_{DNN}(\boldsymbol{m},\theta_{DNN}) = f_{L_{NN}}\left(\ldots f_l\left(\ldots f_1(\boldsymbol{m})\right)\right)$. Specifically, $\boldsymbol{F}_{DNN}(\cdot)$ and $\theta_{DNN}$ represent the DNN-based surrogate model operator and the corresponding trainable parameters, respectively; $f_l(\cdot)$ denote the nonlinear transformation function of the $l$-th layer, and $L_{NN}$ indicates the total number of neural network layers. In DNN model construction, various neural network layers can yield diverse DNN models, resulting in different predictive performances (Lecun et al., 2015). For the DNN models adopted in this study, the involved neural network types are the fully connected layer, the convolutional layer, and the residual block layer.

In fully connected layers, both input and output layers are in vector forms. Assume $X_{\text{input}} \in \mathbb{R}^{n \times 1}$ is the input vector and $X_{\text{output}} \in \mathbb{R}^{m \times 1}$ is the output vector of the $l$-th fully connected layer $f_l(\cdot)$. The transformation in this fully connected layer is expressed as:

$$X_{\text{output}} = f_l(X_{\text{input}}, \theta_l) = f_{\sigma\text{-}l}(W_{DNN}X_{\text{input}} + B_{DNN}) \tag{7}$$

where $f_{\sigma\text{-}l}(\cdot)$ is a non-linear active function; $W_{DNN} \in \mathbb{R}^{m \times n}$ is the weight matrix; and $B_{DNN} \in \mathbb{R}^{m \times 1}$ is the bias vector.

In a convolutional layer, both the input and output are in matrix forms. A convolutional layer transfers information through sparse connections by several convolution kernels, essentially small matrices. The mathematical formula of a convolutional layer is as follows (Wang et al., 2019; Jardani et al., 2022):

$$h_{u,v}^q(x_{u,v}) = f_{\sigma\text{-}l}\left(\sum_{i=1}^{k_i'}\sum_{j=1}^{k_j'} w_{i,j}^q x_{u+i,v+j} + b\right) \tag{8}$$

where $x_{u,v}$ is the pixel value at position $(u, v)$ of the input matrix; $h_{u,v}^q(x_{u,v})$ is the output feature calculated by employing the $q$th ($q=1,\ldots,N_{\text{out}}$) convolutional kernel filter $w^q \in \mathbb{R}^{k_i' \times k_j'}$. In a convolutional layer with $N_{\text{out}}$ filters, the output matrix contains $N_{\text{out}}$ feature layers. The output size ($S_{\text{out}}$) of each convolutional layer is determined by the input size ($S_{\text{in}}$) and the hyperparameters (i.e., zero padding $p$, kernel size $k'$ and stride $s$). A pooling layer is often used after a convolutional layer to remove redundant information from the extracted features and improve the efficiency of model training (Chen et al., 2021).

The residual block is a fundamental component of residual networks (ResNets), designed primarily to mitigate the vanishing and exploding gradient problems commonly encountered during DNN training. A residual block learns a residual mapping defined as:

$$R(X_{\text{input}}, \theta_R) = H(X_{\text{input}}) - T(X_{\text{input}}) \tag{9}$$

where $\theta_R$ represent the trainable parameters of a residual block; $R(\cdot)$ is the residual function; $H(\cdot)$ denotes the target mapping of the residual block aims to approximate; and $T(\cdot)$ is chosen as an identity transformation (i.e., $T(X_{\text{input}}) = X_{\text{input}}$), or another suitable transformation depending on network architecture; The output of the residual block is computed as:

$$X_{\text{output}} = f_{\sigma\text{-}R}(R(X_{\text{input}}, \theta_R) + T(X_{\text{input}})) \tag{10}$$

where $f_{\sigma\text{-}R}(\cdot)$ is the activation function of ReLU. Such design ensures that the output of the residual block at least approximates the input, effectively addressing the vanishing gradient problem. When stacking multiple residual blocks, the relationship between the $L$-th residual block in a deeper layer and the $l$-th residual block is expressed as follows (He et al., 2016):

$$X_{\text{output}(L)} = X_{\text{input}(l)} + \sum_{i=l}^{L-1} R(X_{\text{output}(i)}, \theta_{R(i)}) \tag{11}$$

where $X_{\text{input}(i)}$ and $\theta_{R(i)}$ denote the input data and trainable parameters of the $i$-th residual block, respectively; $X_{\text{output}(L)}$ represents the output from the $L$-th residual block. According to the chain rule in derivatives, the gradient of the loss function $J_{Res}$ with respect to $X_{\text{input}(l)}$ can be given by:

$$\frac{\partial \boldsymbol{J}_{Res}}{\partial \boldsymbol{X}_{\text{input}(l)}} = \frac{\partial \boldsymbol{J}_{Res}}{\partial \boldsymbol{X}_{\text{output}(L)}}\left(1 + \frac{\partial}{\partial \boldsymbol{X}_{\text{input}(l)}}\sum_{i=l}^{L-1}R\big(\boldsymbol{X}_{\text{output}(i)}, \theta_{R(i)}\big)\right) \tag{12}$$

This formulation highlights two key properties of the residual network. First, the gradient does not vanish during network training processes because the term $\frac{\partial}{\partial \boldsymbol{X}_{\text{input}(l)}}\sum_{i=l}^{L-1}F\big(\boldsymbol{X}_{\text{input}(i)}, \theta_{R(i)}\big)$ is never equal to -1. Second, the gradient of the deepest residual block $\frac{\partial \boldsymbol{J}_{Res}}{\partial \boldsymbol{X}_{\text{output}(L)}}$ can directly affect all preceding layers, ensuring effective transmission of gradients throughout the network (Chang et al., 2022).

Based on the three unique network layer structures described above, the FC-DNN, LeNet and ResNet models are constructed. The FC-DNN of this study is constructed using fully connected layers, and each hidden layer consists of 512 neurons. The activation function for the output layer is Sigmoid to constrain outputs within the range of 0 to 1. For hidden layers, the Swish activation function is adopted due to its smooth form with non-monotonic and continuously differentiable properties, which helps improve the DNN training procedures (Elfwing et al., 2018). The performance of the FC-DNN is sensitive to the number of hidden layers, whose optimal value is determined based on specific case studies presented in the application section. For the LeNet and ResNet models, when dealing with low-dimensional scenarios, an initial processing module consisting of a fully connected layer followed by a reshaping operation is added to convert the input vector into a fixed-size matrix. In contrast, for high-dimensional parameter scenarios, the discrete grid matrix of the parameter field is directly input into the CNN architecture (see Figure 1 (b)). Specifically, LeNet consists of two convolutional blocks and two fully connected layers. Each convolutional block consists of a convolutional layer followed by a max-pooling layer. The fully connected layers have 1024 and 512 neurons, respectively. ResNet consists of four stages and two different Res blocks are adopted. The first stage includes two residual units without down-sampling, while the remaining three stages each have one residual unit with down-sampling and one residual unit without down-sampling. Activation functions in all layers are Rectified Linear Units (ReLUs), except for the output layer, where Sigmoid activation is used. Detailed architecture information for LeNet and ResNet is provided in Figure S1 and Figure S2, respectively.

(2) DNN model training

The surrogate models are trained by minimizing the difference between the predicted outputs $\hat{\mathbf{y}}_i = \boldsymbol{F}_{DNN}(\boldsymbol{m}_i, \theta_{DNN})$ and the corresponding numerical model outputs $\mathbf{y}_i$ in training datasets ($i=1,\ldots,N_{samples}$). Following prior researches (Mo et al., 2019, 2020; Chen et al., 2021), the loss function is formulated with L1 norm constraints:

$$\theta^*_{DNN} = \text{argmin}\ \frac{1}{N_{samples}}\sum_{i=1}^{N_{samples}}|\boldsymbol{F}_{DNN}(\boldsymbol{m}_i, \theta_{DNN})\text{-}y_i| + \frac{w_d}{2}\theta^T_{DNN}\theta_{DNN} \tag{13}$$

where $w_d$ is the weight decay to avoid overfitting, referred to as the regularization coefficient. This study implemented the DNN models using PyTorch (https://pytorch.org/). The neural network weights were initialized using the default initialization method of PyTorch and optimized using the stochastic gradient descent method via the Adam algorithm.

## 2.2 Dimensionality reduction methods

### 2.2.1 Karhunen-Loève Expansion for Gaussian random field

Let $Y_G(s) \sim N(\mu_G(s), C(\cdot,\cdot))$ represent a Gaussian random field, where $\mu_G$ denotes the mean of the random field, and $C(\cdot,\cdot)$ represents the exponential covariance function between two arbitrary spatial points $s=(s_x, s_y)$ and $s'=(s'_x, s'_y)$. The covariance function for these two spatial locations is given by:

$$C\left(s, s'\right) = \sigma_G^2 \exp\left(-\sqrt{\left(\frac{s_x - s'_x}{\lambda_x}\right)^2 + \left(\frac{s_y - s'_y}{\lambda_y}\right)^2}\right),$$

(14)

where $\sigma_G^2$ is the variance, $\lambda_x$ and $\lambda_y$ are the correlation lengths along the $x$ and $y$ directions, respectively. Since the covariance matrix is symmetric and positive definite, the exponential covariance function in equation (14) can be decomposed into an eigenvalue-eigenfunction representation. By solving the second-kind Fredholm integral equation and performing eigenvalue decomposition, the Gaussian random field can be expressed through the Karhunen-Loève Expansion (KLE) as follows:

$$Y_G(s) = \mu_G(s) + \sum_{i=1}^{\infty} z_i \sqrt{\lambda_i} \phi_i(s)$$

(15)

where $z_i$ represents a random variable following a Gaussian distribution of $z_i \sim N(0,1)$, also known as a KL term; $\phi_i(s)$ and $\lambda_i$ denote the eigenfunction and eigenvalue, respectively. For discretized numerical models, the index $i$ takes values from 1 to $n$, which represents the number of discrete grid points (i.e., in equation (15), $\infty$ is replaced by $n$). Dimensionality reduction via KLE is achieved through a truncated expansion (Loève, 1955; Zhang and Lu, 2004; Mariethoz and Caers, 2014).

### 2.2.2 Octave Convolution Adversarial Autoencoder for Non-Gaussian random field

The Octave Convolutional Adversarial Autoencoder (OCAAE) is a generative machine learning approach that combines the Variational Autoencoder (VAE) with adversarial learning, leveraging Octave Convolution Neural Networks (Zhan et al., 2021). It consists of three main components: an encoder, a decoder, and a discriminator. The encoder maps a high-dimensional parameter field $m_i$ to a low-dimensional latent vector $z_i$. The distribution of the latent vectors $\{z_1, \ldots, z_N\}$, obtained by mapping the $N$ prior model parameter samples $\{m_1, \ldots, m_N\}$, is denoted as $z \sim q(z)$. Specifically, the encoder outputs two low-dimensional vectors: the mean vector $\mu_z$ and the log-variance vector $\ln(\sigma_z^2)$ of the latent vector $z$. Then, a vector $z'$ is randomly drawn from a standard normal distribution $N(0, I)$, and the latent vector is produced as $z = \mu_z + \sigma_z \times z'$. The decoder reconstructs the high-dimensional parameter field $\tilde{m}$ by taking the latent vector $z$ as input. The discriminator enforces adversarial training, ensuring that the encoded latent vector distribution $z \sim q(z)$ approximates a prior Gaussian distribution $z \sim p(z)$. It receives input from the latent vectors generated by the encoder $z \sim q(z)$ or from the prior distribution $z \sim p(z)$, and discriminates which distribution the input latent vector originates from.

This adversarial framework enhances the generative capability and ensures smooth transitions between different field realizations. In the adversarial autoencoder method, the encoder $G(\cdot)$ (which also acts as the generator of the adversarial network), decoder, and discriminator $D(\cdot)$ are trained jointly in two phases during each iteration: the reconstruction phase and the regularization phase.

In the reconstruction phase, the encoder and decoder are updated using the following loss function:

$$\mathcal{L}_{ED} = \frac{1}{N}\sum_{i=1}^{N} \|\boldsymbol{m}_i - \widetilde{\boldsymbol{m}}_i\|_1 - w_{adv}\left(\frac{1}{N}\sum_{i=1}^{N} \log\{D[G(\boldsymbol{m}_i)]\}\right) \tag{16}$$

where $w_{adv}$ is a weight balancing the reconstruction and adversarial losses (set to 0.01 in this study); $\widetilde{\boldsymbol{m}}_i$ is the reconstructed sample of $\boldsymbol{m}_i$; and $N$ is the number of training samples.

In the regularization phase, the discriminator is trained to distinguish real latent vectors from the prior distribution based on the loss function:

$$\mathcal{L}_D = -\frac{1}{N}\sum_{i=1}^{N}\{\log[D(\boldsymbol{z}_i)] + \log[1 - D[G(\boldsymbol{m}_i)]]\} \tag{17}$$

This loss function helps the discriminator distinguish between the latent vector $z_i$ (from the true distribution $p(z)$) and the fake latent vector produced by the encoder $G(\boldsymbol{m}_i)$.

The constraint loss functions in the adversarial autoencoder framework ensure that the reconstructed high-dimensional parameter field $\widetilde{\boldsymbol{m}}$ closely matches the original field $\boldsymbol{m}$, while also making sure that the distribution of the low-dimensional latent vector $z$ approximates a predefined standard normal distribution $p(z)$. After finishing the training process, it is possible to sample from the low-dimensional space of $p(z)$ and use the decoder to generate corresponding high-dimensional parameter fields. Then, the high-dimensional parameter field can be reconstructed by indirectly estimating the low-dimensional latent vectors (Makhzani et al., 2015; Mo et al., 2020).

## 2.3 Optimization algorithms

### 2.3.1 Metaheuristic algorithms

The four metaheuristic algorithms used in this paper essentially update model parameters through distinct heuristic stochastic search strategies. Specifically, particle swarm optimization (PSO) updates model parameters $\boldsymbol{m}$ based on the personal best position of particles and the global best position of the swarm (Eberhart and Kennedy, 1995). Genetic algorithm (GA) encodes the initial model parameter samples using binary encoding, then iteratively updates them through crossover (combining portions of encoded solutions to generate new candidate solutions), mutation (randomly altering encoded information to introduce diversity), and selection (choosing candidate solutions based on objective function evaluations) (Holland John, 1975). Differential Evolution (DE) initializes a population of real-valued parameter vectors and iteratively updates them through differential mutation (generating trial solutions based on vector differences among population members), crossover (probabilistically combining components from original and mutated vectors), and greedy selection (retaining

solutions with better objective function values) (Storn and Price, 1997; Tran et al., 2022). Simulated Annealing (SA) starts from a random initial solution and iteratively explores neighbouring solutions, accepting them probabilistically based on the Metropolis criterion, while gradually decreasing temperature parameter until convergence (Metropolis et al., 1953; Kirkpatrick et al., 1983).

A common characteristic of all the methods described above is that each iterative update of model parameters requires multiple evaluations of the objective function, and sufficient iterations are necessary to balance local exploitation and global exploration. Detailed implementation procedures and theoretical foundations of these methods are provided in the supplementary materials. The metaheuristic algorithms used in this study were implemented using the open-source Python package scikit-opt (https://scikit-opt.github.io/ ).

**2.3.2 TNNA algorithm**

The TNNA algorithm aims to obtain a reverse network $F_{\text{Reverse}}(\cdot)$ that maps the observation vector to model parameters, as shown in equation (18).

$$m = F_{Reverse}(\mathbf{y}_{\text{obs}}, \theta_{Reverse}) \tag{18}$$

where $\theta_{Reverse}$ are the trainable parameters of $F_{\text{Reverse}}$. Since $m$ also serves as the input to the established surrogate model
$F_{Forward}(\cdot)$, by substituting the parameter $m$ in the inversion objective function of equation (2) with the expression from equation (18), we obtain the objective function constraint for $\theta_{Reverse}$ (i.e., the loss function for training $F_{\text{Reverse}}$):

$$\theta_{Reverse}^* = argmin \sum_{i=1}^{N_{obs}} \frac{1}{\sigma_i} \Big[ \mathbf{y}_{\text{obs}}[i] - F_{Forward}(F_{Reverse}(\mathbf{y}_{\text{obs}}, \theta_{Reverse}))[i] \Big]^2 \tag{19}$$

After obtain the optimal trainable parameters $\theta_{Reverse}^*$ through backpropagation based stochastic gradient descent within the pytorch framework, the final inversion results for the model parameters can be computed by $m^* = F_{Reverse}(\mathbf{y}_{\text{obs}}, \theta_{Reverse}^*)$. The
required training data here are the normalized observation data. Specifically, the reverse network for this study is designed using an FC-DNN with three hidden layers, each containing 512 neurons.

During reverse network training processes, each iteration of updating the trainable parameters $\theta$  involves two main steps: First, the observation vector $\mathbf{y}_{\text{obs}}$ is input into the reverse network $F_{\text{Reverse}}$ to obtain the parameter prediction vetcor $\tilde{m}$. Next, the predicted parameter $\tilde{m}$ is input into the forward network $F_{\text{Forward}}$ to generate corresponding forward prediction results.
Subsequently, the trainable parameters $\theta_{Reverse}$ of the reverse network are updated through standard DNN model training based on the error feedback from the loss function in equation (19). This process demonstrates that $F_{\text{Reverse}}$ and $F_{\text{Forward}}$ are integrated through a tandem connection, which is why this method is named TNNA. Upon completing the training of $F_{\text{Reverse}}$, the final optimal parameters are predicted by inputting observation data into $F_{\text{Reverse}}$. Further details on TNNA can be found in (Chen et al., 2021).

In the above process, each backpropagation step involves only a single forward calculation of the loss function. After establishing the computational graph, gradients of the trainable parameters $\theta_{Reverse}$ are computed through backpropagation

combined with automatic differentiation. These gradients are then used to update the trainable parameters $\theta_{Reverse}$. Thus, only one forward simulation is executed during each epoch of the reverse network $\boldsymbol{F}_{Reverse}$ training procedure. This presents a marked computational advantage of TNNA compared to the four selected metaheuristic algorithms, which require numerous forward simulations for parameter updates at each iteration.

## 3. Case Study

This study considers three synthetic cases based on previous research, covering different model sizes and hydraulic gradient combinations (Jose et al., 2004; Zhang et al., 2018; Mo et al., 2019) to evaluate the performance of the TNNA algorithm against conventional metaheuristic algorithms. Both Case 1 and Case 2 are approximately tens of meters in size, with simulation time measured in days. Their hydraulic gradients are 0.05 and 0.1, respectively. These scenarios are typically found in large sand tank experiments, aquifers with natural slopes, or in-situ experimental areas where flow conditions are enhanced through pumping wells. Case 3 simulates contaminant plume migration, has a size of approximately one kilometre, and simulation time measured in years. It uses a hydraulic gradient of 0.00625, representing a smaller natural gradient typically found in alluvial aquifers. Regarding the differences in heterogeneity conditions among these cases, Case 1 features a low-dimensional zoned permeability field scenario; Case 2 involves a high-dimensional Gaussian random permeability field parameterized via the Karhunen-Loève expansion (KLE); and Case 3 uses a high-dimensional non-Gaussian binary random permeability field parameterized by a decoder trained with OCAAE. The numerical models of the three cases are established using TOUGHREACT, which employs an integral finite difference method with sequential iteration procedures and adaptive time stepping to solve the flow and transport equations. Dispersion effects are inherently incorporated through molecular diffusion and numerical dispersion induced by upstream weighting and grid discretization (Xu et al., 2011).

After developing numerical models for the three scenarios, we first evaluate four surrogate models in Case 1, and the optimal surrogate model will be integrated into the inversion framework. Subsequently, hypothetical observation scenarios are used to systematically compare the inversion accuracy of TNNA against four metaheuristic algorithms across the three cases. The observation data (hydraulic heads and solute concentrations) for model parameter inversion are generated by adding Gaussian noise perturbations to the numerical model simulation results. Specifically, observational noise is introduced by multiplying the min-max normalized simulated data by a random noise factor $\epsilon \sim N(1, \sigma^2)$, where $\sigma$ represents the ratio of observational noise to the observed values. In this study, we conduct a comparative analysis of inversion performance across the three cases under a noise level of $\sigma=0.01$. Additionally, our previous study (Chen et al., 2021) examined the effects of higher observational noise levels ($\sigma=0.05$ and 0.1) and real-world noise conditions on inversion accuracy in low-dimensional parameter scenarios. To further investigate the impact of increased observational noise on inversion performance in high-dimensional parameter scenarios, we conducted an extended analysis on Case 3—the most complex scenario—by increasing the noise level to 10% ($\sigma=0.1$). This analysis also provides insights into the stability of the TNNA algorithm when integrated

with a generative machine learning-based inversion framework for high-dimensional parameter estimation. The details of these three cases are provided in Sections 3.1 to 3.3.

**3.1 Case 1: Low-dimensional zoned permeability field scenario**

As shown in Figure 2, the numerical model for the low-dimensional scenario focuses on conservative solute transport within a zoned permeability field. The model domain is a two-dimensional rectangular area measuring 10m×20m. The left and right boundaries are Dirichlet boundary conditions, with a hydraulic head difference of 1 m. The heterogeneous permeability is divided into eight homogeneous permeability zones, denoted as $k_1$ to $k_8$. The prior range for these eight permeabilities is from $1×10^{-12}$ to $9.9×10^{-12}$ m$^2$. The contaminant source is located at the left boundary with a fixed release concentration ranging from $1×10^{-3}$ to 1 mol/L. The simulation area is uniformly discretized into 3,200 (40×80) cells, and the simulation time is set to 20 days.

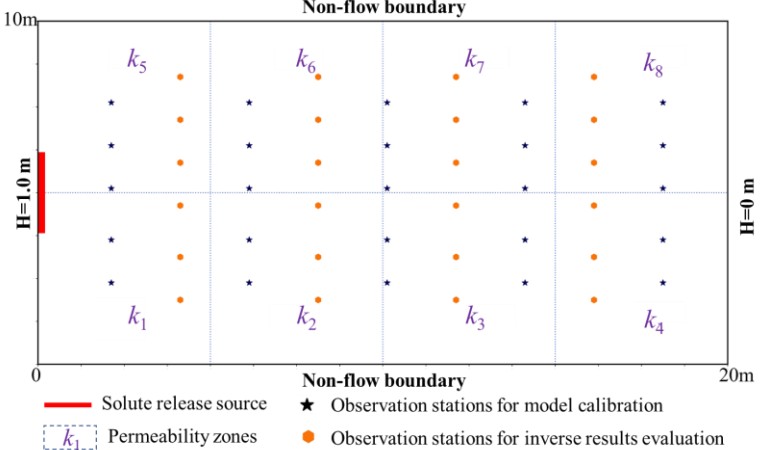

**Figure 2.** Flow domain of the solute transport model for the low-dimensional scenario.

According to these model conditions, there are nine model parameters to be estimated: eight permeability parameters ($k_1$ to $k_8$) and the source release concentration. As shown in Figure 2, these parameters will be estimated using the observation data of hydraulic heads and solute concentrations collected from 25 locations, denoted by black pentagrams. Additionally, observation data from another 24 locations, denoted by orange hexagons and not included in the calibration process, will be used to evaluate the prediction accuracy of the calibrated numerical model.

**3.2 Case 2: High-dimensional gaussian random permeability field scenario**

The numerical model for the high-dimensional scenario features a domain size of 10m×10m, with impervious upper and lower boundaries and constant head boundaries at the left (1m) and right (0m) sides. The domain is discretized into 4,096 (64×64) cells. The log-permeability field follows a Gaussian distribution, and the permeability value of the $i$-th mesh is defined as follows:

$$k_i = \alpha_i k_{ref} \qquad (20)$$

where $k_{ref}$ is the reference permeability, set to $2 \times 10^{-13} \text{m}^2$. The heterogeneity of $k_i$ is controlled by the modifier $\alpha_i$. The geostatistical parameters for this Gaussian field are: $m = 0$, $\sigma_G^2 = 2$, and $\lambda_x = \lambda_y = 2.5$ m. Under this heterogeneous condition, 100 KLE terms are used to preserve more than 92.67% of the field variance. Consequently, estimating the permeability field is equivalent to identifying these 100 KLE terms.

The observational data used for inverse modeling include hydraulic heads from a stationary flow field and solute concentrations measured every two days over 40 days, starting from the 2nd day to the 40th day (day: $t = 2i$, $i = 1, \ldots, 20$). It should be noted that in high-dimensional parameter scenarios, the increased degrees of freedom typically result in greater parameter uncertainty. Insufficient observational information may fail to effectively constrain parameter estimation, resulting in potential uncertainty and equifinality (Mclaughlin and Townley, 1996; Zhang et al., 2015; Cao et al., 2025). Therefore, this study

includes actual permeability values at observed locations as regularization constraints to mitigate inversion errors arising from equifinality. Since identical regularization conditions are uniformly applied across all algorithms, introducing these constraints ensures the stability and robustness of the inversion outcomes without affecting the inherent performance characteristics of the five optimization algorithms compared in this study.

        As the degrees of freedom significantly increase in high-dimensional models, the influence of observation data on

inversion results becomes increasingly significant. Five scenarios with different monitoring networks are considered to comprehensively evaluate the performance of different inversion algorithms using various observations. Figure 3 displays the monitoring station locations for each scenario.

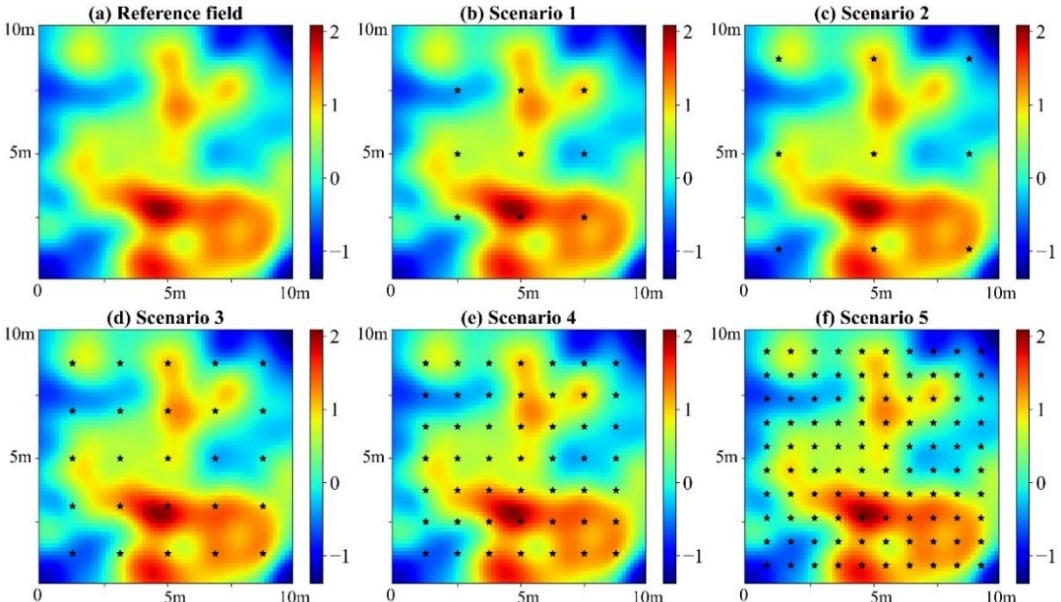

**Figure 3.** The reference log-permeability field and locations of observation stations for five scenarios. The observation stations

are represented by black pentagrams.

### 3.3 Case 3: High-dimensional non-gaussian random permeability field scenario

This case focuses on an estimation of a binary non-Gaussian permeability field. The numerical model features a domain size of 800m×800m, with impervious upper and lower boundaries and constant head boundaries at the left (5m) and right (0m) sides. The domain is discretized into 6400 (80×80) cells. The permeability field is a channelized random field composed of two lithofacies, with permeability values of $1.0×10^{-13}m^2$ and $5.46×10^{-12}m^2$ for the two media, respectively. The reference field (Figure 4b) is generated from a training image (Figure 4a) using the direct sampling (DS) method proposed by Mariethoz et al. (2010). The contaminant release source is located at the entire left boundary, with a concentration of 1 mol/L. The observational data used for inversion are generated through numerical simulation, including steady-state hydraulic head data and solute concentration data at 12 time points (from 2 years to 24 years, with 2-year intervals). This case focus on a high-dimensional binary inverse problem aimed at identifying the lithofacies type of each discrete grid cell within the domain. Note that the permeability values of the two lithofacies are fixed in this case.

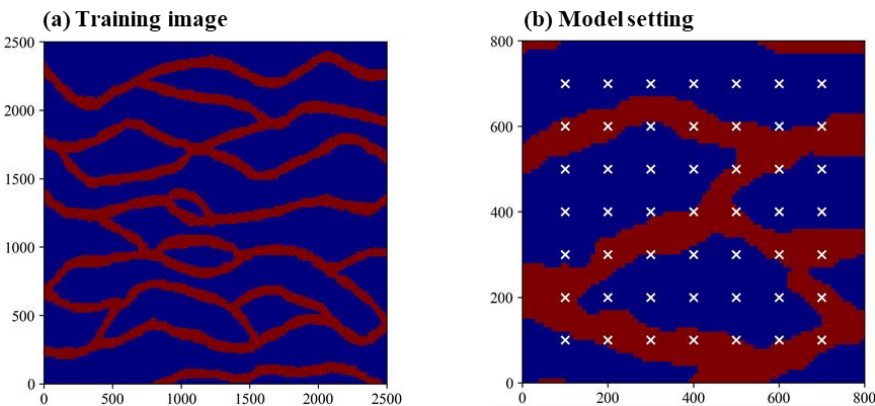

**Figure 4.** (a) The training image used to generate random realizations of permeability field; (b) The reference field of the synthetic case (white symbols indicate observation locations).

To achieve low-dimensional representation of permeability fields, a training dataset comprising 2000 stochastic realizations is generated using multi-point statistics (MPS). Then, an Octave convolution-based Adversarial Autoencoder (OCAAE) is developed, where the decoder network learns a nonlinear mapping from 100-dimensional Gaussian latent vectors to 6400-dimensional binary non-Gaussian permeability fields. Thus, the non-Gaussian permeability field is indirectly reconstructed by estimating the 100-dimensional latent vector.

## 4. Results and discussion

### 4.1 Surrogate model evaluations

Surrogate models were first compared using the Case 1 with low-dimensional parameter. For this scenario, the input parameters for the surrogate models consist of a 9-dimensional vector, including 8 permeability parameters and the

contaminant source release concentration. The output consists of the simulated hydraulic heads and solute concentrations at 25 observation points. Four training datasets $D_{train}=\{M_{train}, Y_{train}\}$ with 200, 500, 1000, and 2000 samples (represented as $D_{train\text{-}200}$, $D_{train\text{-}500}$, $D_{train\text{-}1000}$ and $D_{train\text{-}2000}$, respectively) and a testing dataset $D_{test}=\{M_{test}, Y_{test}\}$ with 100 samples (represented as $D_{test\text{-}100}$) are prepared. These datasets were generated using Latin hypercube sampling (LHS) and numerical simulations. The predictive accuracy of surrogate models was quantitatively evaluated using root mean square error ($RMSE$) and determination coefficient ($R^2$) metrics (Chen et al., 2022).

For solute transport inverse modeling problems, it is crucial to consider observations of both hydraulic heads and solute concentrations simultaneously. Therefore, the surrogate model within an inversion framework should have accurate predictive capabilities for hydraulic heads and solute concentrations. This study calculates $RMSE$ and $R^2$ values separately for hydraulic heads, solute concentrations, and all model response data, resulting in the following evaluation criteria: $RMSE_{ALL}$ and $R^2_{ALL}$ for overall data, $RMSE_H$ and $R^2_H$ for hydraulic heads, and $RMSE_C$ and $R^2_C$ for solute concentrations. Additionally, it should be noted that the above $RMSE$ and $R^2$ metrics are computed based on the normalized hydraulic head and solute concentration data.

Figure 5 and Figure 6 display the $RMSE$ and $R^2$ values of each surrogate model, and Figures S3 to S6 in the supplementary material present the pairwise comparison results. The optimal values for $C$, $\sigma$, and $\varepsilon$ in the MSVR method are provided in Table S1. For the FC-DNN, the optimal number of hidden layers was separately determined for each of the four datasets. The candidate range for the number was set from 1 to 7. According to the $RMSE_{All}$ and $R^2_{All}$ values in Table S2 and Table S3, optimal number of hidden layers for in the FC-DNN for $D_{train\text{-}200}$, $D_{train\text{-}500}$, $D_{train\text{-}1000}$ and $D_{train\text{-}2000}$ are 2, 4, 3, and 3, respectively.

According to the performance criteria in Figure 5 and Figure 6, the prediction accuracy of each surrogate model significantly improves with an increasing number of training samples. Based on $RMSE_{All}$ and $R^2_{All}$ values, their performance ranks as follows: ResNet, LeNet, FC-DNN, and MSVR. The MSVR method accurately predicts hydraulic heads but performs the worst in predicting solute concentration. Training MSVR with the four prepared datasets, the $RMSE_H$ values are below 0.02, and $R^2_H$ values are near 1. Notably, with a training sample size of 200, the prediction accuracy of MSVR for hydraulic heads is higher than that of FC-DNN and LeNet, as indicated by their $RMSE_H$ and $R^2_H$ values, closely matching that of ResNet. However, when using 200 training samples, the $RMSE_C$ value for MSVR exceeds 0.08, and the $R^2_C$ value falls below 0.85. Even with a dataset size of 2000, the enhancement in the MSVR-based surrogate model is limited, as the $RMSE_C$ value remains around 0.05, and the $R^2_C$ value stays below 0.95. FC-DNN demonstrates a significant advantage over MSVR in predicting solute concentration, particularly with larger training sample sizes of 1000 or 2000. However, there are still some obvious biases between some surrogate modeling results and their numerical modeling results (see Figure S2(d)). When adopting CNN-based surrogate models (LeNet and ResNet), the prediction accuracy for solute concentrations significantly improves (see Figure 5(b) and Figure 6(b)). With training datasets of 2000 samples, LeNet and ResNet achieve $RMSE$ values below 0.02 and $R^2$ values close to 1. It is worth noting that the ResNet performs well even with smaller sample sizes. For example, with 200 training samples, the $RMSE_C$ and $R^2_C$ values for LeNet are around 0.06 and 0.9, respectively, while these criteria values for

ResNet are around 0.04 and 0.95 (see Figure 5(b) and Figure 6(b)). As the number of training samples increases, the advantages of ResNet become more apparent. According to Figure S4(d), when the training sample size reaches 2000, the prediction results of ResNet are closely consistent with the numerical simulation results for both hydraulic heads and solute concentrations.

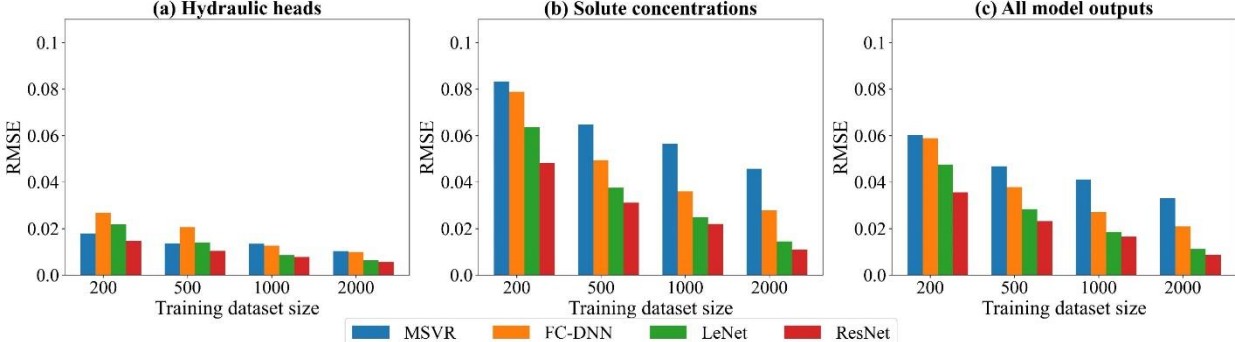

**Figure 5.** The *RMSE* results of surrogate model predictions. Plots (a) to (c) show respectively the *RMSE* values of hydraulic heads, solute concentrations and all model outputs.

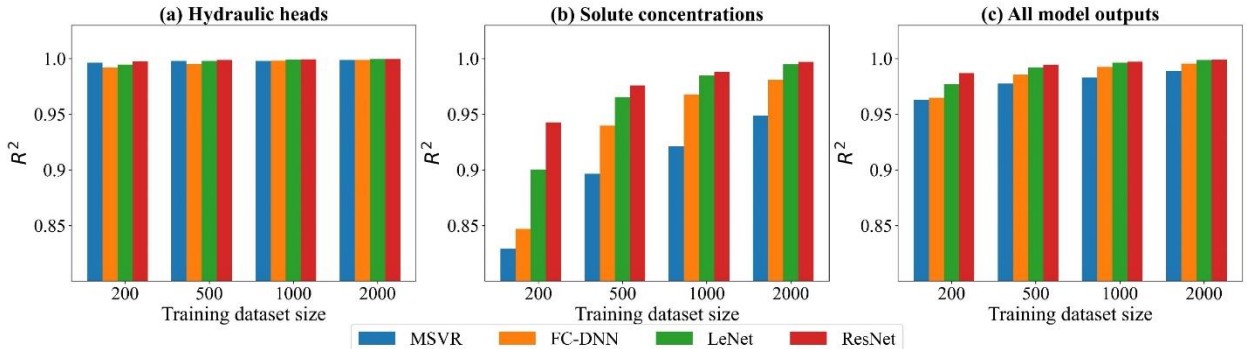

**Figure 6.** The $R^2$ results of surrogate model predictions. Plots (a) to (c) show respectively the $R^2$ values of hydraulic heads,
solute concentrations and all model outputs.

The comparison results of the surrogate models reflect a trend of enhanced robustness attributable to advancements in machine learning methodologies. Different machine learning approaches employ distinct strategies for achieving nonlinear mappings in developing surrogate models. Generally, deeper or larger models contain more trainable parameters, resulting in higher degrees of freedom to capture more robust nonlinear relationships. The essence of machine learning development lies
in addressing the challenge of training these complex DNNs. Current state-of-the-art machine learning techniques have demonstrated proficiency in training each of the four selected surrogate modeling methods. With sufficient training samples, a surrogate model of greater complexity exhibits enhanced capability in representing higher levels of non-linearity (Lecun et al., 2015; He et al., 2016). This also explains why, despite having a sufficient number of training samples, the improvement in prediction accuracy of the MSVR for solute concentration is limited. In CNNs, sparse connections and weight-sharing in
convolutional layers reduce redundant weight parameters in DNNs, enhancing the feature extraction of hidden layers. Consequently, LeNet demonstrates better performance than FC-DNN. The ResNet, which employs residual blocks in

conjunction with convolutional layers, effectively addresses the issues of gradient vanishing and exploding, making the successful training of deeper CNNs possible.

According to Chen et al. (2021), a more globally accurate surrogate model can enhance the performance of TNNA inversion results. Thus, we selected the ResNet trained with 2000 samples for the subsequent inversion procedure. In the low-dimensional scenario, its *RMSE* values for hydraulic head and solute concentration data are less than 0.02, with $R^2$ values greater than 0.99. We further extended the ResNet for the surrogate model construction of both Gaussian and non-Gaussian random field scenarios. In the two high-dimensional scenarios, the input parameters for the surrogate models are single-channel matrix data representing the heterogeneous parameter field, while the output consists of vector formed by flattening the multi-channel matrix data, representing the simulated hydraulic heads and solute concentrations at predefined time steps within the simulation domain. The training and testing datasets for these two case scenarios consist of 2000 and 500 samples, respectively. The *RMSE* values for hydraulic head and solute concentration data range from approximately 0.01 to 0.03, and the $R^2$ values exceed 0.99, as shown in Table 1. This level of accuracy indicates that the surrogate model meets the predictive accuracy requirements for inversion simulations in both of the designed Gaussian and non-Gaussian random field cases.

**Table 1.** The *RMSE* and $R^2$ values for surrogate model predictions in designed five high-dimensional scenarios.

| | *RMSE* | | | $R^2$ | | |
|---|---|---|---|---|---|---|
| | $RMSE_H$ | $RMSE_C$ | $RMSE_{All}$ | $R_H^2$ | $R_C^2$ | $R_{All}^2$ |
| Gaussian Scenario-1 | 0.0108 | 0.0174 | 0.0172 | 0.9990 | 0.9980 | 0.9982 |
| Gaussian Scenario-2 | 0.0102 | 0.0138 | 0.0136 | 0.9995 | 0.9989 | 0.9990 |
| Gaussian Scenario-3 | 0.0120 | 0.0165 | 0.0163 | 0.9991 | 0.9981 | 0.9983 |
| Gaussian Scenario-4 | 0.0123 | 0.0161 | 0.0159 | 0.9990 | 0.9984 | 0.9985 |
| Gaussian Scenario-5 | 0.0137 | 0.0156 | 0.0155 | 0.9989 | 0.9985 | 0.9986 |
| Non-Gaussian Scenario | 0.0181 | 0.0280 | 0.0273 | 0.9952 | 0.9931 | 0.9932 |

## 4.2 Parameter inversion method comparison results

### 4.2.1 Inversion results of the low-dimensional parameter scenario

For the low-dimensional parameter scenario, the performance of optimization algorithms is thoroughly evaluated across 100 parameter scenarios using the Monte Carlo strategy. The observation data for these scenarios are derived from the testing dataset after adding multiplicative Gaussian random noise $\epsilon \sim N(1, 0.01)$. The population sizes of GA, DE, and PSO, along with the chain length in SA, are set in four distinct scenarios: 20, 40, 60 and 80 (these population size or chain length values are represented as $N_{PC}$ in subsequent discussions). These settings determine the number of forward modeling calls required for each iteration, significantly influencing the convergence rate and computational efficiency of optimization procedures. Maximum iterations for these four metaheuristic algorithms are set to 200. The learning rate, epoch number and weight decay for the TNNA algorithm are set to $6 \times 10^{-5}$, 1000, and $1 \times 10^{-6}$, respectively.

The performance of the five optimization algorithms is evaluated according to three aspects: average convergence efficiency and accuracy in inversion procedures, predictive accuracy of calibration models for hydraulic heads and solute

concentrations, and statistical analysis of the estimated errors for each model parameter. Figure 7 presents the logarithmic average convergence curves (i.e., $\log_{10}$ of the average objective value during inversion iterations) of four metaheuristic algorithms and the TNNA algorithm throughout 100 parameter scenarios. Specifically, sub-figures (a) to (d) represent the $N_{PC}$ values for metaheuristic algorithms set at 20, 40, 60, and 80, respectively. These figures clearly illustrate the average convergence speed and accuracy of five optimization algorithms. Figure 8 displays the comparison between simulated and observed values across all 100 parameter scenarios for both calibration and spatial predictive evaluation. Sub-figures (a) and (b) illustrate the comparative prediction fit at the 25 observation locations used for model calibration, whereas sub-figures (c) and (d) display the comparative prediction fit at the 24 independent observation locations. In this figure, distinct symbols are used to represent the five optimization algorithms. It should be noted that the $N_{PC}$ values for the four metaheuristic algorithms are uniformly set to 80 during this comparison. Figure 9 illustrates the probability density curves of the estimation errors for nine model parameters across 100 parameter scenarios, with different colours representing the five optimization algorithms.

The results in Figure 7 demonstrate that the TNNA algorithm achieves the best convergence accuracy, with its convergence logarithmic objective function value (i.e., approximately -4.4) being smaller than those of the other four metaheuristic algorithms across these $N_{PC}$ settings. The influence of $N_{PC}$ on the convergence speeds of these four metaheuristic algorithms is not significant, exhibiting a distinct transition from rapid to slower convergence around the 75th iteration. As $N_{PC}$ increased from 20 to 80, each metaheuristic algorithm showed distinct improvements in the accuracy of the final objective function. The DE algorithm showed the least improvement in final convergence accuracy as the $N_{PC}$ value increased from 20 to 80, with the logarithmic value of its objective function dropping from just above -4.0 to slightly below -4.0. The SA algorithm also showed limited improvement, with its logarithmic average convergence value increasing from around -4.1 at $N_{PC}$=20 to slightly below -4.3 at $N_{PC}$=80, close to that of the TNNA algorithm. Among the four metaheuristic algorithms, SA exhibited the highest average convergence accuracy. Contrary to the SA and DE algorithms, the PSO and GA algorithms significantly enhanced average convergence accuracy as $N_{PC}$ increased. Specifically, as $N_{PC}$ increased from 20 to 80, the logarithmic convergence values of PSO and GA decreased by more than 0.5. While increasing $N_{PC}$ values may help metaheuristic algorithms reduce the gap in average convergence accuracy compared to the TNNA algorithm, larger $N_{PC}$ settings also require additional computational burdens. The above results indicate that the TNNA algorithm has a significant efficiency advantage over the four metaheuristic algorithms in parameter optimization. For instance, when conducting optimization procedure based on scikit-opt, the DE algorithm requires 32,000 forward model realizations (80×2×200) when $N_{PC}$ is set to 80, while the other three metaheuristic algorithms (PSO, GA, and SA) each require 16,000 realizations (80×200). In significant contrast, the TNNA algorithm requires only one forward model realization per iteration, resulting in 200 realizations. These comparisons illustrated that the TNNA method is more effective than the other four metaheuristic algorithms in achieving robust convergence results.

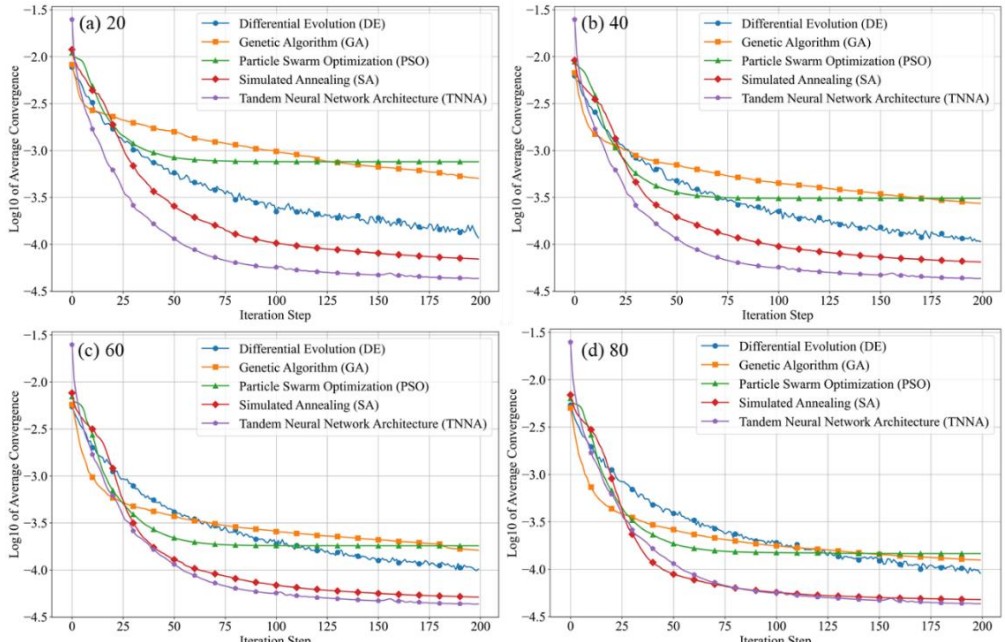

**Figure 7.** Comparative convergence trends ($\log_{10}$ of the average objective value) of five optimization algorithms on 100 parameter scenarios. Plot (a) to (d) compare the four metaheuristic algorithms and TNNA under $N_{PC}$=20, 40, 60, and 80, respectively; TNNA was executed only once on the same 100 parameter scenarios, and its curve is identical across all plots; Markers indicate convergence values every 10 iterations.

The results presented in Figure 8 indicate that, among the five optimization algorithms, the TNNA algorithm achieves the smallest *RMSE* values and $R^2$ values closest to 1.0 for both hydraulic heads and solute concentration during model calibration and spatial predictive evaluation. Furthermore, the distribution of comparison points demonstrates that the modeling results obtained from both calibration and independent prediction using the TNNA algorithm match the observed values more accurately than those of the other four metaheuristic algorithms, particularly for solute concentrations. Among the four metaheuristic algorithms, SA and DE outperform GA and PSO regarding *RMSE* and $R^2$ values. During model calibration and predictive evaluation, PSO exhibits the worst predictive accuracy, recording the highest *RMSE* and $R^2$ values for both hydraulic heads and solute concentrations. It is noteworthy that the *RMSE* and $R^2$ values for SA during hydraulic head calibration are 0.0085 and 0.9992, respectively, while those for DE during solute concentration calibration are 0.0112 and 0.9969. These values are almost equal to those of the TNNA algorithm. The robustness of an inversion algorithm is determined by its accuracy in both calibration and predictive evaluation for hydraulic heads and solute concentrations. However, DE and SA demonstrate appropriate calibration accuracy only for one of the two simulation components. Overall, the TNNA algorithm provides more robust model calibration and predictive evaluation results than the other four metaheuristic algorithms.

Figure 9 indicates that the estimated error distributions for the nine model parameters derived from the TNNA algorithm are more concentrated than those obtained from the four metaheuristic algorithms. The mean estimated error values for the nine numerical model parameters using the TNNA algorithm are also the lowest. These results highlight the high accuracy and

reliability of the TNNA inversion algorithm. Among the four metaheuristic algorithms, DE and SA outperform GA and PSO. This is because the probability density curves of estimation errors for the nine parameters using DE and SA are more concentrated around zero, with mean values lower than those of GA and PSO. The DE algorithm shows a more concentrated distribution around zero for the overall estimation errors of parameters $k_1$ to $k_8$. In contrast, the SA reveals reduced estimation errors for the $C_0$ parameter in most cases, ranking just behind the TNNA algorithm. GA outperforms PSO in estimation

accuracy for seven of the nine model parameters, with PSO matching its probability density curves to that of GA only for parameters $k_2$ and $k_4$. As a whole, the statistical results of the estimated model parameter errors illustrate that the machine learning-based TNNA algorithm exhibits enhanced inversion performance compared to the four metaheuristic optimization algorithms. However, the findings also reveal that none of the five algorithms consistently offers completely reliable inversion solutions across all scenarios. For example, the TNNA algorithm, despite its generally better performance, demonstrates

estimation errors as high as 0.4 for parameters $k_4$ and $k_6$ in some scenarios. Such results are likely because the provided observational data cannot ensure equifinality in some scenarios. In these cases, it is essential to introduce additional regularization constraints to attenuate the equifinality (Wang and Chen, 2013; Arsenault and Brissette, 2014). These findings emphasize the importance of employing the Monte Carlo method in comparative studies of inversion algorithms to ensure comprehensive evaluations and avoid misleading conclusions.

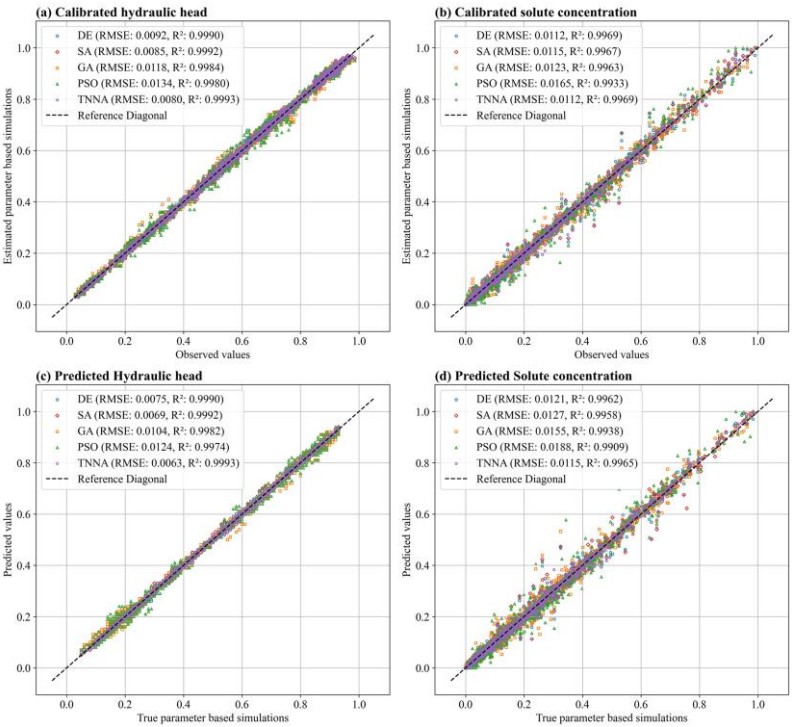

**Figure 8.** Comparison of predictive accuracy for hydraulic heads and solute concentrations simulated using parameters estimated by the four metaheuristic inversion algorithms (DE, SA, GA, PSO) and the TNNA method. Sub-figures (a) and (b) show predictive comparisons at the 25 observation locations used for model calibration; sub-figures (c) and (d) show predictive comparisons at the other 24 independent observation locations.

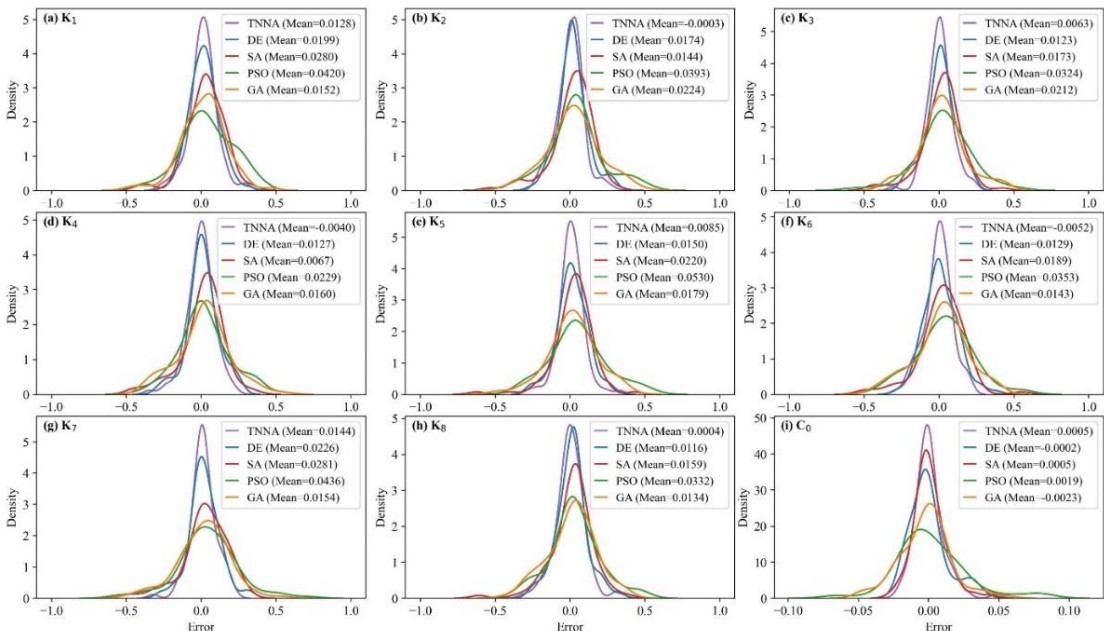

**Figure 9.** Probability density curves of estimation errors for nine model parameters using five optimization methods. Each curve represents the distribution of estimation errors across 100 parameter scenarios, with their mean error values indicated in the legends.

The above comparison results indicated that the machine learning-based TNNA algorithm outperforms the other four metaheuristic algorithms in both inversion accuracy and computational efficiency. The primary advantage of the TNNA algorithm over the four metaheuristic algorithms is its well-defined updating direction of model parameters, guided by the loss function, which serves as the objective function for inverse modeling. Research on machine learning applications indicates that DNNs can approximate continuous functions by adjusting weights and biases (Lecun et al., 2015; Goodfellow et al., , 2016). The TNNA algorithm leverages this capability by transforming the model parameter inversion issue into the training of a reverse network to achieve reverse mappings. By establishing a loss function based on inversion constraints from the Bayesian theorem, the TNNA algorithm ensures that training the reverse network brings each parameter update closer to the optimal solution during each epoch, thereby improving accuracy and convergence speed. In contrast, the four metaheuristic algorithms require numerous forward simulations for each parameter update. The optimization direction for model parameters is determined by evaluating the objective function. This process is governed by the exploration and exploitation strategies inherent in metaheuristic algorithms. However, these approaches introduce randomness in the direction of model parameter updates, making it challenging to ensure that updates move towards the direction of fastest convergence under specific hyperparameter settings. This also explains why the TNNA algorithm can update model parameters more efficiently and achieve higher convergence accuracy despite requiring only one forward realization in each training epoch.

**4.2.2 Inversion results of the high-dimensional Gaussian scenario**

For estimating the permeability field under five designed observational scenarios, the iteration number for the four metaheuristic algorithms was set at 200, with $N_{PC}$ values of 100, 500, and 1000. The learning rate and weight decay for training reverse networks within the TNNA framework were set to $1\times10^{-3}$ and $1\times10^{-4}$, respectively.

      Figure 10 and Figure 11 illustrate the log-permeability field estimation results and error distributions for the four metaheuristic algorithms and the TNNA algorithm under the most densely observed scenario (i.e., Scenario 5). The

620 corresponding results for Scenarios 1-4 are presented in Figure S7-S14. Figure 12 compares the *RMSE* values for the log-permeability fields estimated by the four metaheuristic algorithms and the TNNA algorithm across all five scenarios. These detailed *RMSE* values can be found in Table 2 (Scenario 5) and Table S4 (Scenarios 1-4). For Scenario 5, the accuracy of permeability estimations by each metaheuristic algorithm improves as the $N_{PC}$ value increases (see Figure 10 and Table 2). Notably, the GA achieves the best results with an $N_{PC}$ of 1000, recording an *RMSE* of 0.1057. The DE and SA algorithms yield

their most accurate permeability estimations with *RMSE* values of 0.1597 ($N_{PC}$=100) and 0.1549 ($N_{PC}$=1000), respectively. The PSO method is the least effective, achieving an *RMSE* of 0.3334 at $N_{PC}$ =1000. As shown in Figure 11 and Table 2, the TNNA algorithm provides inversion results with an *RMSE* of 0.1063 after training the reverse network for 200 epochs. This suggests that the TNNA algorithm can estimate high-dimensional permeability fields with accuracy comparable to that of the GA method ($N_{PC}$=1000) with significantly fewer forward model realizations (200 compared to 200,000), reducing the

computational burden by 99.9% and improving inversion efficiency by a factor of 1000. Increasing the training epochs of the reverse network to 1000 further reduces the *RMSE* of the TNNA method to 0.0595, demonstrating its advantages over the four metaheuristic algorithms in this scenario. Across all scenarios, the accuracy of the estimated permeability fields correlates positively with the density of observation wells, and estimation errors are generally higher in areas not covered by monitoring wells (see Figure S7-S14). Figure 12 further demonstrates that the *RMSE* values for permeability estimation using the TNNA

algorithm are consistently lower than those of the four metaheuristic algorithms across Scenarios 1-4, indicating that the TNNA algorithm exhibits greater robustness compared to the metaheuristic algorithms in all five scenarios.

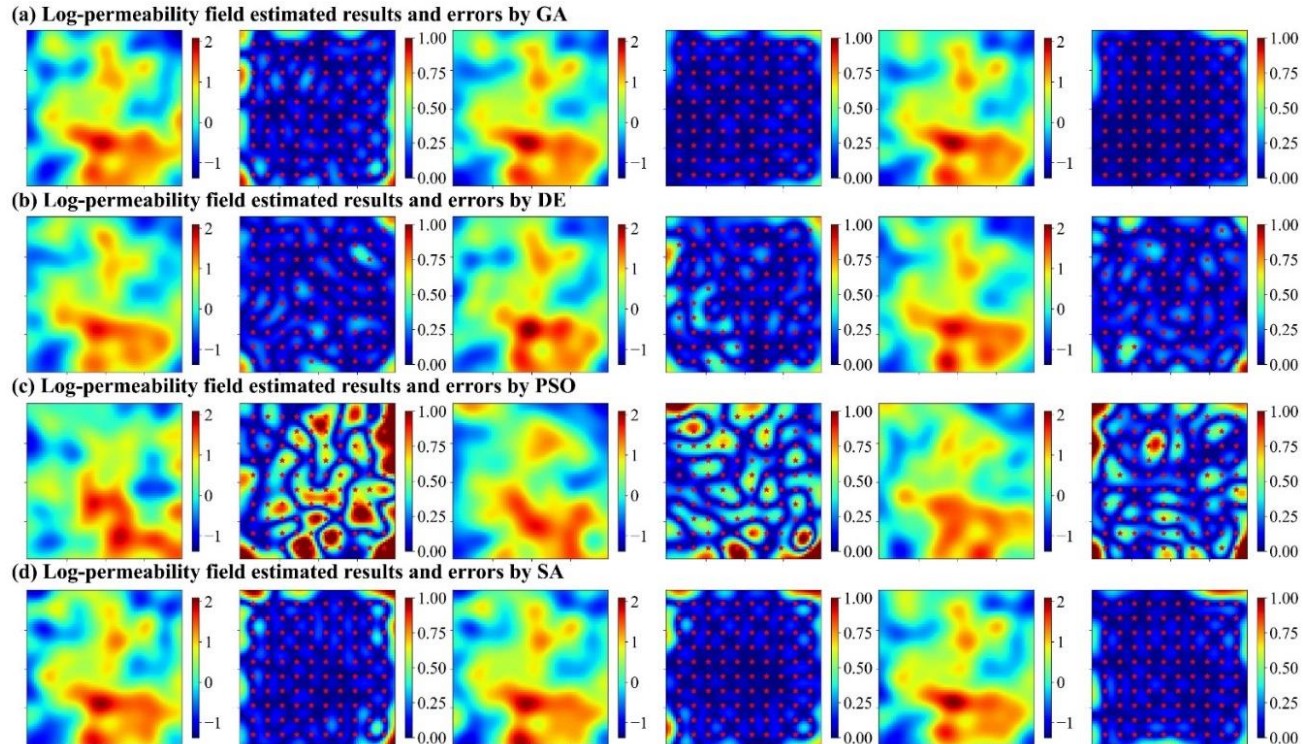

**Figure 10.** Spatial distributions of log-permeability field estimation results (row 1, 3, and 5 for $N_{PC}$=100, 500, and 1000, respectively) and absolute errors (row 2, 4, and 6 for $N_{PC}$=100, 500, and 1000, respectively) for Scenario 5, achieved by four metaheuristic algorithms (plots (a) to (d) correspond to GA, DE, PSO and SA, respectively).

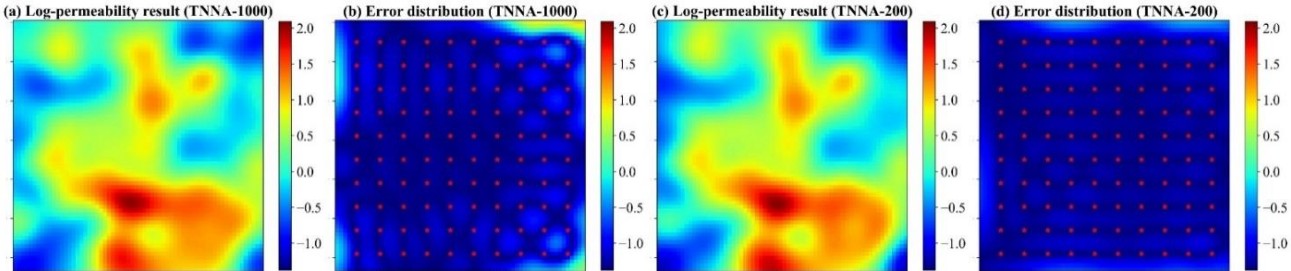

**Figure 11.** Spatial distributions log-permeability field estimation results and absolute errors for Scenario 5, achieved by the TNNA. Plots (a) and (c) show the log-permeability fields estimated using 1000 (TNNA-1000) and 200 (TNNA-200) training samples, respectively; plots (b) and (d) present the corresponding absolute error distributions.

**Table 2.** *RMSE* values of estimated log-permeability fields for the four metaheuristic algorithms and the TNNA algorithm under Scenario 5.

| | Metaheuristic algorithms | | | | TNNA | |
|---|---|---|---|---|---|---|
| | **GA** | **DE** | **PSO** | **SA** | | |
| $N_{PC}$=100 | 0.1940 | 0.1597 | 0.5399 | 0.2071 | epoch=200 | 0.1063 |
| $N_{PC}$=500 | 0.1142 | 0.1904 | 0.3810 | 0.1781 | epoch=1000 | 0.0595 |
| $N_{PC}$=1000 | 0.1057 | 0.1748 | 0.3334 | 0.1549 | | |

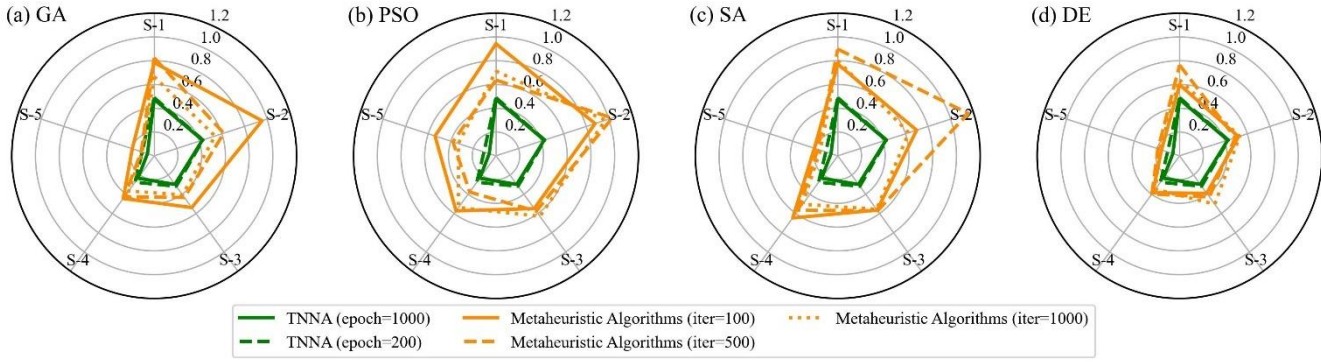

**Figure 12.** Comparison of *RMSE* in estimating log-permeability fields using four metaheuristic algorithms and the TNNA algorithm across five scenarios (S-1 to S-5).

To evaluate the predictive performance of the numerical model calibrated by various inversion methods, simulations of hydraulic heads and solute concentrations were conducted over 60 days, starting on the 2nd day with recordings every two days, using the permeability fields with the lowest *RMSE* values identified by each inversion method. Observation data from the 2nd day to the 40th day were used for model calibration, while additional data from the 42nd to the 60th day were employed to evaluate the future predictions of the calibrated numerical models. The *RMSE* values for the calibrated hydraulic heads and time series solute concentrations are presented in Table 3 and Figure 13. Figure 14 displays the spatial distribution of the calibrated numerical simulation results and errors for hydraulic heads and solute concentration simulation results at three specific times (t=4th, 20th, and 52nd days). Results for the entire 60-day period are presented in Figure S15-S44.

According to Figure 14(a), the calibrated simulation errors for hydraulic heads did not exceed 0.02 meters for the TNNA method and three of the four considered metaheuristic algorithms, except PSO method, which exhibited hydraulic head errors larger than 0.06 meters in certain areas. Among the four metaheuristic algorithms, the GA method achieved the lowest *RMSE* in hydraulic head simulations, with a value of $7.4837 \times 10^{-4}$. For solute concentrations, the GA algorithm consistently has the highest prediction accuracy among the metaheuristic algorithms, with *RMSE* values generally around 0.005 (Figure 13). The TNNA algorithm achieved a similar level of accuracy to GA in the calibrated numerical model predictions. Specifically, during the initial 10 days and from the 41st day to the 60th day, the TNNA algorithm showed slightly higher prediction accuracy than the GA-calibrated model. However, during the intermediate period from the 10th day to the 40th day, the GA-calibrated model had a slight advantage over the TNNA algorithm. The normalized absolute errors in the solute transport simulation results obtained using the TNNA algorithm remained consistently below 0.02 throughout the simulation period (Figure 14(b to c)). These results indicate that in high-dimensional settings, the TNNA algorithm provides inversion outcomes that enable the calibrated model to deliver simulation results comparable to those of the best-performing metaheuristic algorithm. Overall, the TNNA method also demonstrates advantages over the four metaheuristic optimization algorithms in the designed high-dimensional scenarios, excelling in both inversion efficiency and accuracy.

**Table 3.** *RMSE* values of calibrated hydraulic heads for the four metaheuristic algorithms and the TNNA algorithm.

| | TNNA | DE | GA | PSO | SA |
|---|---|---|---|---|---|
| RMSE | $6.8537 \times 10^{-4}$ | $1.2181 \times 10^{-3}$ | $7.4837 \times 10^{-4}$ | $2.1683 \times 10^{-3}$ | $1.0316 \times 10^{-3}$ |

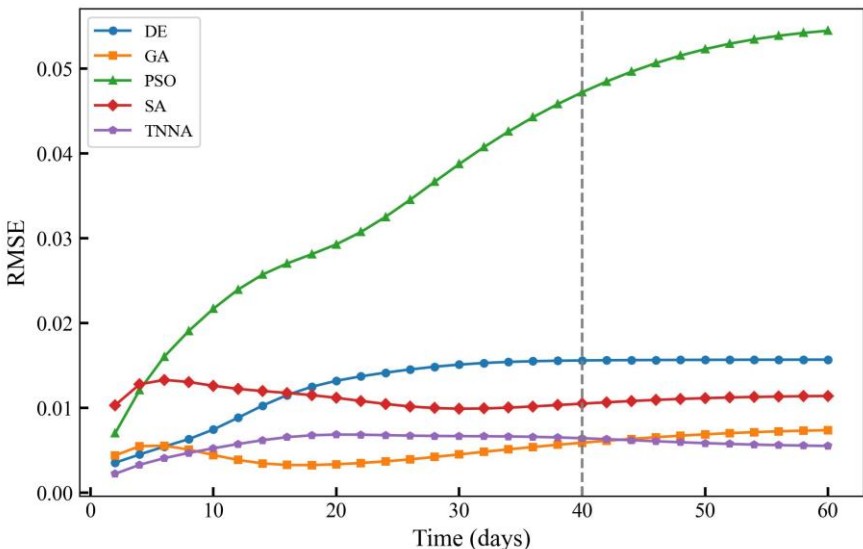

**Figure 13.** *RMSE* **values of calibrated solute concentrations over 60 days for the four metaheuristic algorithms and the TNNA algorithm.**

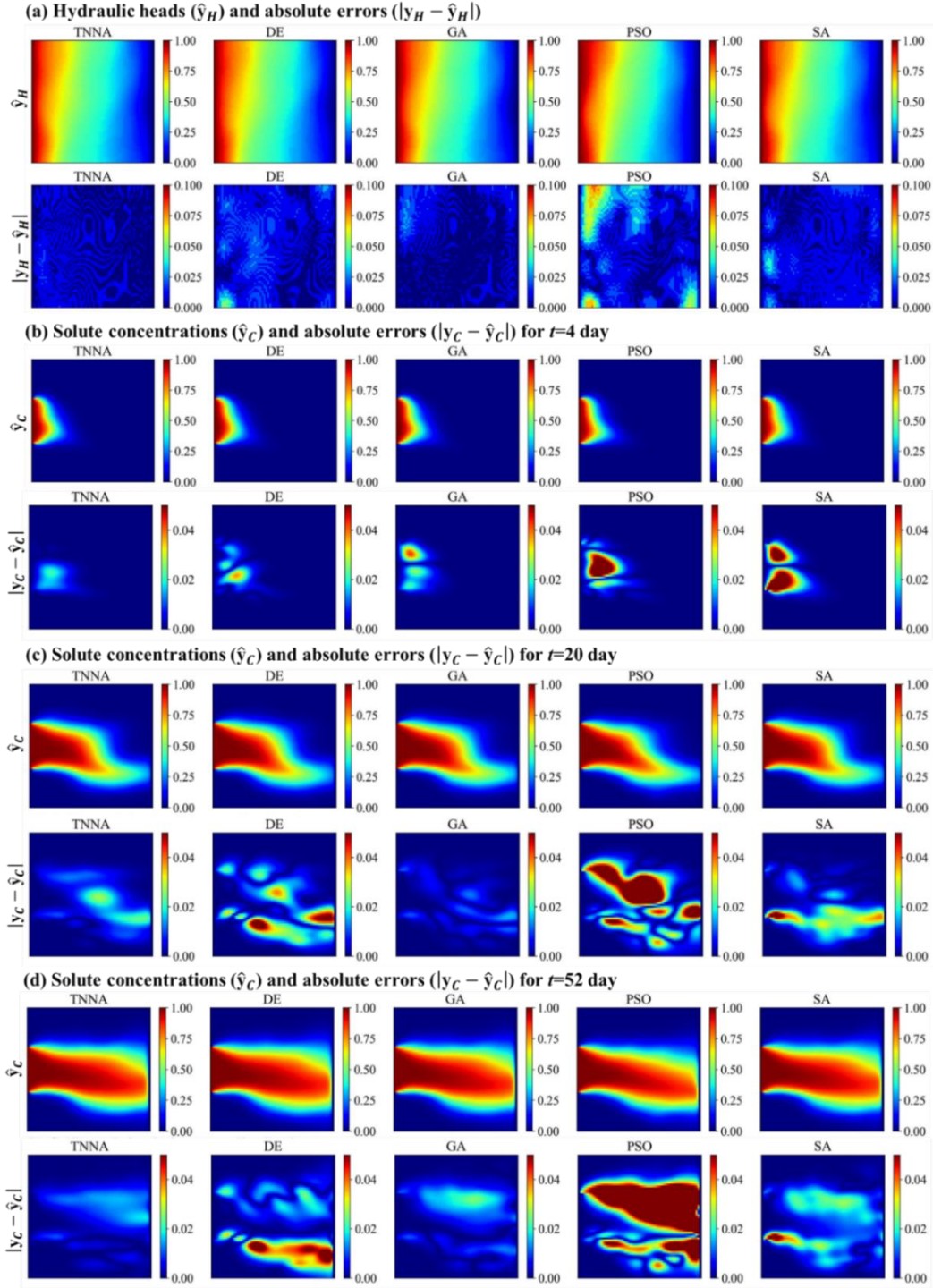

**Figure 14. Spatial distributions of calibrated numerical simulation results and absolute errors for hydraulic heads and solute concentrations at three dynamic times (t=4, 20, and 50 day) using the TNNA algorithm and four metaheuristic algorithms.**

## 4.2.3 Inversion results of the high-dimensional non-Gaussian scenario

In this scenario, the iteration number for the four metaheuristic algorithms was set at 200, with $N_{PC}$ values of 1000. For the TNNA method, the reverse network is trained for 1000 epochs. Thus, each metaheuristic algorithm spent 100 times more forward model evaluations than the TNNA algorithm. Figure 14 and Figure 15 show the permeability fields estimated by the five optimization algorithms and their error distributions compared to the true field (i.e., the error fields). Figure 16(a) and Figure 17(a) present the comparison between calibrated simulations and hydraulic head observations, as well as solute concentration observations. Figure 16(b) and Figure 17(b) compare the solute concentration simulations for the $26^{th}$, $28^{th}$, and $30^{th}$ years based on the estimated parameter field and the designed true field.

According to Figures 15 and 16, the binary channel fields reconstructed by each inversion algorithm are highly consistent with their corresponding true fields, with the estimated errors primarily concentrated at the interfaces between high-permeability channels and low-permeability regions. It is found that increasing the observation noise level from 1% to 10% does not lead to noticeable increase in the number of grid cells exhibiting differences between the estimated parameter fields and the true field. One potential reason for this is that the least-squares objective function used in the inversion framework of this study is based on the assumption that the observation noise follows a zero-mean Gaussian distribution. With adequate regularization constraints, such as the dense monitoring network design used in this study, the model responses corresponding to the optimal parameter estimates obtained through global optimization algorithms statistically converge to the mean of the observed data. It can also be evaluated by the calibration simulations. Specifically, the pairwise scatter plots in Figure 17(a) and Figure 18(a) indicate that the calibrated simulation results from different methods are closely distributed around the reference diagonal. This suggests that even with increased observational noise, the inversion-derived calibration results do not exhibit noticeable bias. Furthermore, the predictions based on inversion results remain highly consistent with those of the true permeability field (Figure 17(b) and Figure 18(b)). The $RMSE_{All}$ and $R^2_{All}$ values for the predictions beyond the observational period range from 0.018 to 0.044 and 0.962 to 0.994, respectively. This indicates that even under relatively high Gaussian noise conditions, the nonlinear inversion framework used in this study can reliably reconstruct the non-Gaussian permeability field, ensuring high predictive accuracy. Nevertheless, it is important to note that while the inversion accuracy under a 10% noise level remains comparable to that in the 1% noise scenario, increasing observational noise inevitably raises the convergence value of the least-squares loss function. This trend is evident from the $RMSE$ values in Figures 17(a) and 18(a). Moreover, since the observational noise here is assumed to follow a Gaussian distribution, real-world scenarios with more complex noise characteristics may further exacerbate equifinality in the inversion results. In such cases, incorporating additional system information as regularization constraints is essential to enhance the robustness of the objective function and mitigate ill-posedness.

Compared to the four metaheuristic algorithms, TNNA demonstrates advantages in computational efficiency and accuracy for non-Gaussian random field inversion. In the low noise level scenario, TNNA achieves an inversion convergence accuracy with an RMSE$_{All}$ of 0.021 and an $R^2_{All}$ of 0.996 (Figure 17(a)). In contrast, the two best-performing metaheuristic methods, GA

and SA, yield RMSE$_{All}$ values of 0.027 and 0.029, with $R^2_{All}$ values of 0.994 and 0.993, respectively (Figure 17(a)). Moreover, TNNA achieves the highest fitting accuracy for predictive results among the five optimization algorithms, with an *RMSE* of 0.018 and an R$^2$ of 0.994 (Figure 17(b)). Even in high-noise scenarios, TNNA continues to exhibit an advantage over the four metaheuristic algorithms in both inversion convergence accuracy (Figure 18(a)) and predictive accuracy (Figure 18(b)). Additionally, considering the number of forward simulation calls required by each inversion algorithm, TNNA proves to be a more efficient approach in this case study.

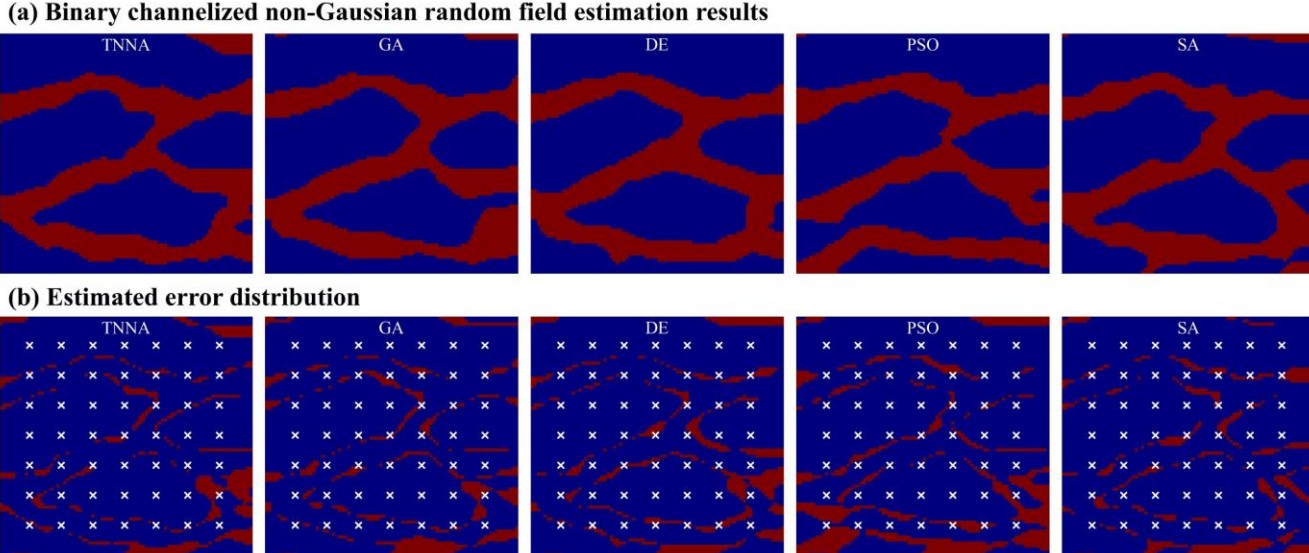

Figure 15. Reconstructed non-Gaussian binary channelized fields and their error distributions (1% observation noise)

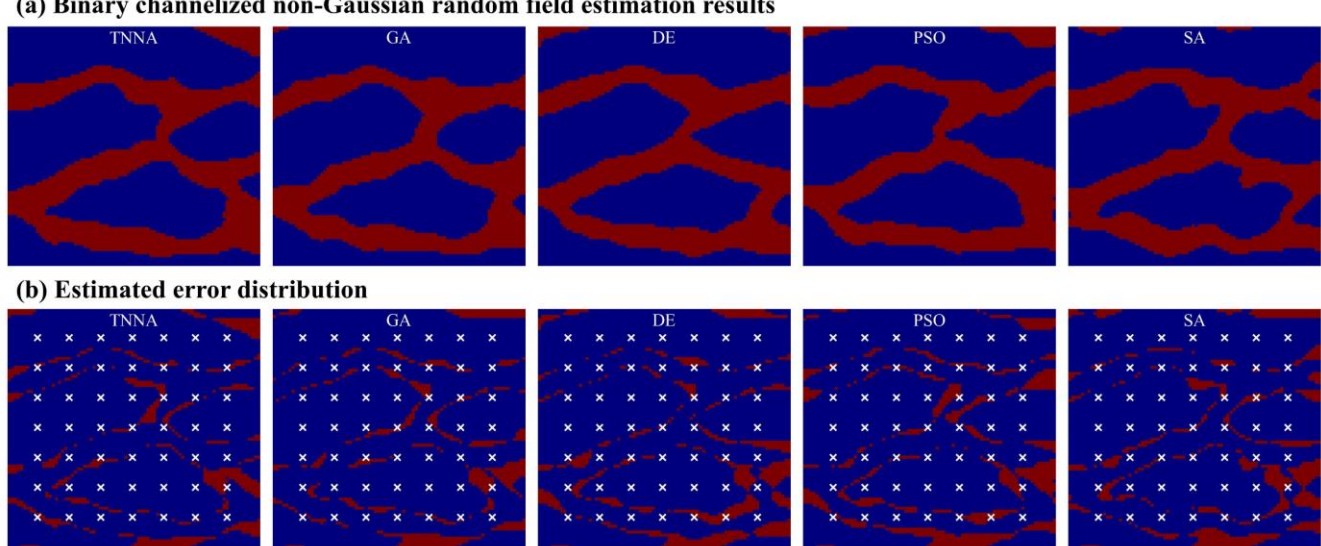

Figure 16. Reconstructed non-Gaussian binary channelized fields and their error distributions (10% observation noise)

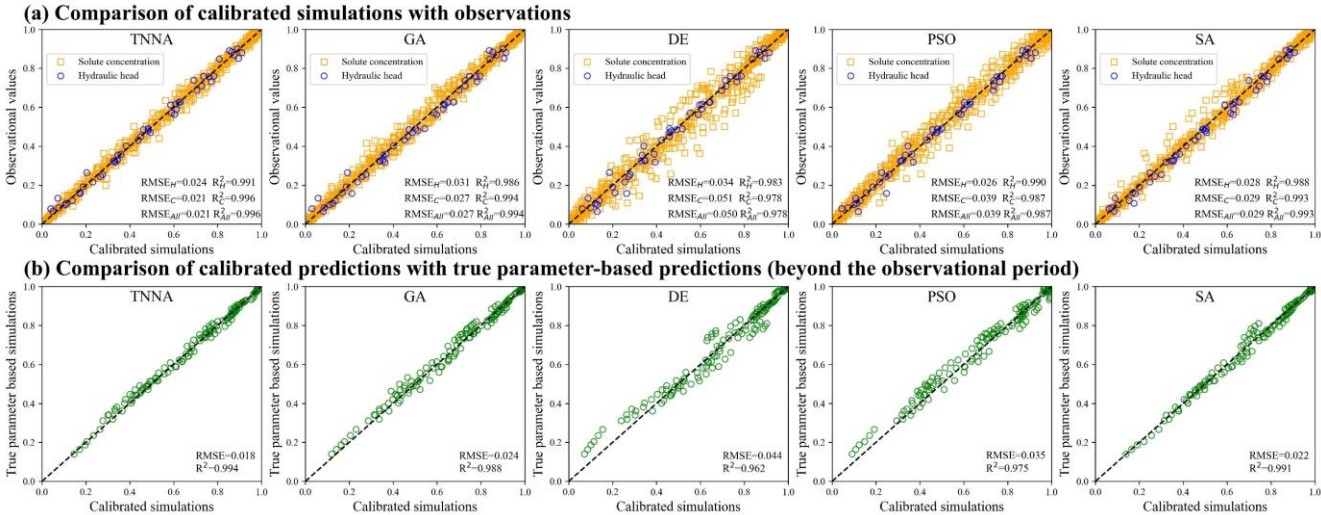

**Figure 17. pair-wise comparison between the calibrated simulation results with the observational data (a); and the true parameter based predictions (1% observation noise).**

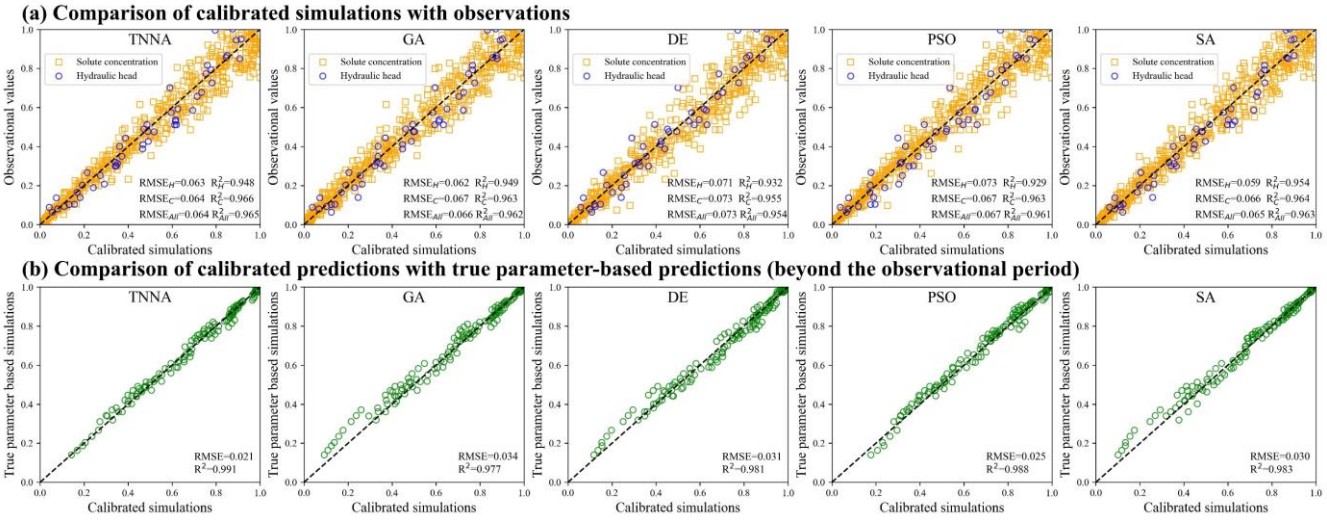

**Figure 18. pair-wise comparison between the calibrated simulation results with the observational data (a); and the true parameter based predictions (10% observation noise).**

### 4.3 Parameter inversion method comparison results

This study evaluates the computational efficiency and inversion reliability of the TNNA algorithm under three different heterogeneous conditions. In optimization-based inversion studies, the primary challenge is to establish nonlinear inversion constraints and design efficient algorithms to find optimal parameter solutions. The main difference between cases lies in how the constraint conditions are formulated, while the optimization algorithm itself remains generally applicable across different optimization tasks if these conditions are properly defined. Therefore, the fundamental challenge in applying well-performing

inversion methods to real-world cases lies in whether robust nonlinear optimization constraints can be effectively established for inversion tasks. Given the complexities of subsurface systems, three key aspects should be considered to extend the TNNA method to real-world applications: 1) Representation of complex heterogeneous model parameter fields; 2) Maximizing the effective observational information while optimizing monitoring costs; and 3) Integrating multi-source data and accounting for uncertainties in model process to better address complex observational noise scenarios and uncertainties in physical

mechanisms. Detailed considerations for these issues are as follows:

- Heterogeneity in aquifer parameter structures: This study developed a dimensionality-reduction framework using the OCAAE for high-dimensional parameter field inversion. Generative machine learning methods (including state-of-the-art variants) also have the potential to characterize more complex non-Gaussian fields. However, obtaining representative parameter field datasets remains challenging in practical research. For instance, spatial variations in non-stationary

stochastic aquifer systems may result in significant discrepancies in geostatistical parameters across sampling windows (Mariethoz and Caers, 2014). Therefore, developing appropriate generator training strategies is essential for these practical scenarios.

- Monitoring network optimization: The inversion performance of the TNNA and four metaheuristic algorithms is evaluated based on a nonlinear optimization model with dense distributed monitoring networks. This monitoring strategy

is commonly employed in the evaluation of inversion algorithms to ensure sufficient observational information, thereby reducing non-uniqueness in parameter inversion results (Bao et al., 2020; Mo et al., 2020; Zhang et al., 2024). Such monitoring strategies for comparing inversion methods also aim to minimize external interferences, ensuring that differences in performance are primarily determined by inversion algorithms themselves. However, the number and locations of monitoring stations are constrained by financial budgets. Thus, optimizing monitoring network design to

minimize monitoring costs without compromising constraint information quality is indispensable for practical applications (Keum et al., 2018; Chen et al., 2022; Cao et al., 2025).

- Considering multi-source data and uncertainties in model processes: This study considers only hydraulic head and solute concentration data, assuming ideal white Gaussian noises. However, in real-world scenarios, observational noise is often more complex and may exhibit non-Gaussian characteristics. For instance, some solute concentrations cannot be

measured in situ, and unavoidable perturbations may be included during sample collection and laboratory analysis. Similarly, hydraulic head data may be influenced by meteorological factors. Moreover, all observational data in this study are constrained by a single predetermined process model. However, if significant uncertainties exist in the actual aquifer model processes or if the conceptual model deviates substantially from real-world conditions, even an advanced optimization algorithm may produce incorrect inversion results. Therefore, it is crucial to integrate multi-source data (e.g.,

geophysical measurements or isotope data) and develop multi-process coupled models to establish more robust inversion frameworks (Dai and Samper, 2006; Botto et al., 2018; Chang and Zhang, 2019). Specifically, parameterizing model process uncertainties to enable the simultaneous identification of both model processes and unknown parameters could be a promising direction for real-world studies.

## 5. Summary and conclusions

This study systematically evaluates the performance of the Tandem Neural Network Architecture (TNNA) in comparison to four widely used metaheuristic algorithms (GA, PSO, DE, and SA) across three inversion frameworks designed for different heterogeneous groundwater conditions. The results demonstrate that TNNA consistently outperforms the four conventional metaheuristic algorithms across the designed scenarios, covering both low-dimensional and high-dimensional cases. It provides more accurate inversion results while significantly reducing computational costs. Moreover, it has been verified that

the TNNA algorithm consistently delivers reliable inversion results with just a single forward simulation per iteration step in scenarios featuring various complex and uncertain model parameters. This characteristic offers a practical approach to balancing exploration and exploitation with a reduced computational burden, contrasting with conventional metaheuristic algorithms that require increasing forward simulations as the inversion problem grows more complex. Furthermore, this study introduces a novel framework that integrates TNNA, along with optimization algorithms, with generative machine learning-

based parameterization methods for dimensionality reduction in complex heterogeneous parameter fields.

In summary, training reverse network through TNNA method provides significant advantages over conventional metaheuristic algorithms. The proposed integrated framework, which combines the TNNA method with dimensionality reduction techniques, further enhances its applicability and demonstrates strong potential for high-dimensional inversion problems. Developing specialized inversion algorithm frameworks based on state-of-the-art machine learning methods tailored

to specific problem scenarios represents a promising research direction. Furthermore, hyperparameters can significantly influence neural network performance in certain scenarios. It is necessary for future research to explore hyperparameter optimization and sensitivity analysis to identify the optimal neural network structures and training strategies, ultimately enhancing model performance across diverse hydrological conditions.

### Competing interests

The contact author has declared that none of the authors has any competing interests.

### Acknowledgments

This work is supported by the Fundamental Research Funds for the Central Universities (XJ2023005201), the National Natural Science Foundation of China (NSFC: 42402241, U2267217, 42141011, and 42002254).

### Data Availability Statement

The data and codes for four surrogate models and five optimization algorithms are available on: https://doi.org/10.5281/zenodo.10499582

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
