# Peer review of "Enhancing Inverse Modeling in Groundwater Systems through Machine Learning: A Comprehensive Comparative Study"

_Hydrology and Earth System Sciences, 2024_

## Author Comment (AC1)

Reviewer 1:
General comments

The preprint deals with the analysis of the performance of different machine learning methods in inverse modelling of groundwater system. It compares the TNNA method described in previous papers by the authors' collective (J. Chen et al., 2021) with other machine learning methods and shows significantly better performance of the TNNA method compared to several other methods.

The scientific contribution of the preprint is fair, but not very good. The paper does not introduce a new method, but presents on two selected academic problems the good performance of a previously described method. It uses appropriate procedures and criteria for comparison and the results are presented in a very clear, lucid and convincing manner.

The study is well constructed and executed; its scientific quality is very good but I have one reservation about the chosen methodology that I will describe in the following paragraph.

Specific comments

My only comment on the methodology used in the preprint, which I consider to be significant, is the failure to include measurement error in the test problems used. The optimization problems are well chosen, but the measurements used to calibrate the parameters (inverse modeling) were not burdened with any random error emulating measurement error.

This may have a significant impact on the applicability of the method. In inverse modelling, in practice, we face two types of problems - the lack of ability to fit the measured data and the so-called overfitting of the data consisting in their too accurate replication by the model (by including the measurement error in the model parameters, i.e. damaging them in terms of the ability of further prediction). If the aim of the study was to show possible applicability of TNNA to solution of inverse models of groundwater problems, this feature of the study does not allow to fulfil the intended objective.

Technical corrections

The language of the preprint is clear, I did not notice any specific errors or typos.

**Response:**
    We sincerely appreciate the reviewer's constructive comments and the time invested in reviewing our manuscript. Your insights are invaluable in helping us improve the quality of our works.
    Regarding the reviewer's concern about the absence of measurement errors in the test problems, we apologize for the lack of clarity in the original manuscript. In fact, we incorporated measurement errors in both test cases by adding Gaussian noise to the normalized numerical simulation results, with a standard deviation of 0.01. This was explicitly stated in Section 3.2, Case 2, where we noted:

*"The standard deviation of Gaussian noise for the normalized observations is set to 0.01."* However, we recognize that the description in Case 1 was insufficiently detailed. To address this, we will revise the manuscript to explicitly clarify that both test cases included Gaussian noise with a standard deviation of 0.01 for the normalized observations.

Additionally, we agree with the reviewer's perspective on the impact of observational noise on inversion results, particularly in real-world scenarios. Nevertheless, these challenges often arise due to insufficiently defined inversion constraints. Because model parameters are strictly updated based on loss functions in available inversion algorithms. As a result, even minor noise perturbations can cause significant errors when constraints are ill-posed and rely on sparse, noisy data. Therefore, in practical applications, both providing effective inversion algorithms and establishing robust inversion constraints using sufficient data or regularized information are important. In this study, we primarily focus on the impact of differences in parameter optimization mechanisms between the machine learning-based TNNA inversion algorithm and traditional metaheuristic algorithms on inversion accuracy and algorithm efficiency. Accordingly, sufficiently well-defined constraints are included in both test cases to avoid inversion errors caused by external factor, which could mislead the performance assessment. When encountering significant observational noise in real-world scenarios, an effective approach is to improve the regularization terms, such as obtaining more measured parameter values, establishing prior information with lower uncertainty, or using other data sources as additional constraints.

We will supplement the manuscript with additional details on the potential challenges of applying the TNNA method to real-world scenarios, including considerations for observational noise and appropriate constraints construction.

Thank you once again for your valuable feedback and thoughtful suggestions.

---

## Author Comment (AC2)

Reviewer 2:

General comments

While the topic covered could be of interest, using a Gaussian covariance (or double exponential, as it is called in the paper) without any nugget effect renders the comparison exercise a purely academic exercise with little or no practical bearing. Besides the fact that the underlying random function model behind the K-L expansion is the multiGaussian one, another decision that is far from reality.

In summary, for this paper to have any practical interest, the comparison exercise should be performed using a clearly non-multiGaussian random function with the kind of spatial variability one is expected to find in the field, not the unrealistic smooth spatial variability induced by a Gaussian covariance function.

**Response:**

We sincerely appreciate the reviewer's insightful comments and the time invested in evaluating our work. We fully recognize the importance of non-Gaussian random fields in real-world applications. In our original manuscript, the two cases were primarily designed to highlight the impact of variations in model parameter dimensions on the comparison of inverse algorithm performance, ignoring their real-world applicability. To address this, we will supplement the manuscript by discussing real-world scenarios that exhibit similar nonlinear characteristics to those in the examples presented, and explaining how the TNNA algorithm can handle inversion problems involving non-Gaussian random fields.

In Case 1, the domain is divided into a finite number of homogeneous zones, each with different parameter values. This parameterization approach is applicable to scenarios where the goal is to uncover the dynamics law of subsurface systems at a macroscopic level. For example, basin-scale groundwater models typically divide aquifers into homogeneous zones based on lithology and geomorphological features. In some field scale cases, divide the domain into homogeneous parameter zones based on weathering extent or fracture density is also a commonly adopted model simplification approach when data is insufficient. Regarding the Gaussian random field assumption in Case 2, when the aquifer consists of a single lithofacies with relatively uniform sedimentary structures, the permeability distribution can be considered to transition smoothly in space. In such scenarios, characterizing the permeability with a Gaussian random field proves to be an effective approach.

However, as the reviewer rightly pointed out, many real-world scenarios require accounting for non-Gaussian random field heterogeneity. For example, in braided bimodal geostatistical models of deltaic aquifers, the heterogeneous parameter fields exhibit spatial variability that does not transition smoothly, making it difficult to approximate using methods like K-L expansion. In such scenarios, both the TNNA algorithm and traditional metaheuristic algorithms may not be directly applicable. A promising approach to address this challenge is to use generative machine learning methods to establish a mapping between the non-Gaussian random field and low-dimensional latent vectors. The inversion of these latent vectors can indirectly reconstruct the non-Gaussian random field. By combining generative machine learning methods with the TNNA algorithm, it becomes possible to solve the inversion problems of non-Gaussian random fields.

We will include this discussion in the revised manuscript. Thank you once again for your

valuable feedback and constructive suggestions.

---

## Author Response (AR1)

**Response to the comments of Editor and Reviewers**

**Manuscript ID:** hess-2024-315

**Title:** Enhancing Inverse Modeling in Groundwater Systems through Machine Learning: A Comprehensive Comparative Study

Dear Editor and Reviewers,

We sincerely appreciate your valuable feedback and the opportunity to revise our manuscript. We have carefully considered each comment and made significant revisions to enhance the methodological depth, case study design, noise robustness analysis, and practical relevance of our work. Below, we provide a detailed, point-by-point response to the comments.

**Comment 1.** A third case study with a different spatial correlation should be added.

**Response:**

We have included an additional case study focusing on a non-Gaussian random field. The details of this new case study are presented in Section 3.3.

Additionally, for the dimensionality reduction of the non-Gaussian random field, the Karhunen-Loève Expansion (KLE) method is no longer applicable. Hence, a new parameterization method based on the Octave Convolution Adversarial Autoencoder (OCAAE) has been introduced in Section 2.2.2.

The surrogate model for this case was constructed using deep residual network (ResNet) with 2000 training samples, and its prediction accuracy is discussed in the last paragraph of Section 4.1. ("*We further extended the ResNet for the surrogate model construction of both Gaussian and non-Gaussian random field scenarios,⋯⋯⋯⋯*")

The inversion results are presented in Section 4.2.3.

**Comment 2.** Noisy data should be used for all the tests, and I would appreciate very much an analysis of the stability of the methods, i.e., which is the behavior of the applied methods, for different noise magnitude.

**Response:**

In the second paragraph of the Section 3, we have supplemented the description of the noise settings for each scenario. Specifically: In the original two cases, we used 1% Gaussian noise, introduced by multiplying the standardized numerical simulation data with a random variable $\varepsilon \sim N(1, 0.01^2)$ to generate noisy observational data. For the newly added Case 3, we have also implemented a 1% noise scenario to ensure consistency.

Additionally, we fully agree with the editor and reviewer's suggestion to explore the impact of different noise magnitudes on the stability of inversion results. To address this, we specifically designed a 10% observational noise scenario for Case 3, as this case involves a high-dimensional parameter space and a more complex heterogeneous condition. Regarding low-dimensional parameter settings, we previously examined the effects of higher observational noise levels ($\sigma = 0.05$ and $0.1$) and real-world noise conditions on inversion accuracy in our earlier study (Chen et al., 2021). Therefore, this study focuses on extending the noise analysis to more complex high-dimensional cases. ("*The observation data for model parameter inversion are generated by adding*

*Gaussian noise perturbations to the numerical model simulation results. Specifically, oobservational noise is introduced by multiplying the simulated data by a random noise factor $\varepsilon \sim N(1, \sigma^2)$, ··········we conducted an extended analysis on Case 3—the most complex scenario—by increasing the noise level to 10% ($\sigma=0.1$)*")

For the results in the high-dimensional noise setting, given that our study assumes Gaussian noise and that observational constraints are sufficiently dense, the impact on the inversion results of model parameters is not significant ("*Nevertheless, it is important to note that while the inversion accuracy under a 10% noise level remains comparable to that in the 1% noise scenario, increasing observational noise inevitably raises the convergence value of the least-squares loss function. ··········In such cases, incorporating additional system information as regularization constraints is essential to enhance the robustness of the objective function and mitigate ill-posedness.*"). However, we acknowledge that real-world scenarios often involve more complex observational noise distributions and practical constraints, such as limited monitoring networks due to cost restrictions. Hence, we have expanded the discussion in the last two paragraphs of Section 4.3, highlighting potential challenges in real-world studies and possible strategies. This issue is also further addressed in our response to Comment 3.

**Comment 3.**   The discussion about the practical relevance of the test cases must be improved. The authors claim that "the model conditions of the two synthetic cases in this study are primarily based on previous studies, as well as large-scale sandbox". Their discussion of the relevance of the results for practical applications is not convincing: the remarks in section 4.3 and in the replies to the reviewers are not well supported from physical arguments and should be deeply revised.

**Response:**

We followed these suggestions to reformat and revise our manuscript deeply. The description of the similarities between our model setup and real-world hydrogeological conditions has been moved to the first paragraph of Section 3, including detail discussions on the newly added Case 3. Specifically, Case 3 now features a lower hydraulic gradient and a model scale of 0.8km. Thus, the three cases collectively cover high, medium, and low hydraulic gradient scenarios, providing a more comprehensive representation of different hydrogeological conditions. ("*This study designed three synthetic cases based on previous research, covering different model scales and hydraulic gradient combinations (Jose et al., 2004; Zhang et al., 2018; Mo et al., 2019) to evaluate the performance of the TNNA algorithm against conventional metaheuristic algorithms. Both Case 1 and Case 2 are small-scale scenarios, with simulation time measured in days.··········ase 3 simulates contaminant plume migration at a sub-regional scale (approximately 1 km), with simulation time measured in years. It uses a hydraulic gradient of 0.00625, representing a smaller natural gradient typically found in plain aquifers.······:*")

In Section 4.3, we have revised and expanded the discussion on practical applications. In the supplemented descriptions, we explain that well-performing optimization algorithms are generally applicable across different optimization tasks, provided that the constraints for inversion are properly defined. However, the primary challenge in real-world studies lies in establishing robust constraint conditions tailored to specific scenarios. We specifically provided a detailed discussion on three potential challenges associated with constructing nonlinear optimization models for real-world applications: 1) Representation of complex heterogeneous model parameter fields; 2)

Maximizing the effective observational information while optimizing monitoring costs; and 3) Integrating multi-source data and accounting for uncertainties in model process to better address complex observational noise scenarios and uncertainties in physical mechanisms.

("*4.3 Parameter inversion method comparison results*

*This study validates the computational efficiency and inversion reliability of the TNNA algorithm under three different heterogeneous conditions. In optimization-based inversion studies, the primary challenge is to establish nonlinear inversion constraints and design efficient algorithms to find optimal parameter solutions·········.*")

---

## Editor Decision (ED1)

1. Lines 13 & 14. Rephrase the sentences, possibly as follows: "Tandem neural network architecture (TNNA) is a machine learning algorithm which as been recently proposed for estimating uncertain parameters with inverse mappings".

2. Lines 22 & 25. Here a percentage of noise is mentioned (1% or 10%), but it is not clearly stated which are the measured quantities and what is used as reference value.

3. Lines 31 to 36. Recent publications only have been considered. However, these concepts are well-established since a long time and can be considered text-book material. Moreover, I wonder whether the papers referenced for inverse modeling are the most relevant ones. Many other review papers on inverse problems in hydrology are available and should be considered (e.g., 10.1016/0022-1694(87)90207-1, 10.2136/sssabookser5.4.c40, 10.1016/S0167-5648(04)80146-1, 10.2166/nh.2007.024, 10.1029/96WR00160, 10.1007/BF01547729, 10.1016/0309-1708(91)90039-Q, 10.3390/hydrology11110189, 10.1029/WR022i002p00095, 10.1002/hyp.3360060305, and many others).

4. Line 39. I would not use "deterministic" to characterize Bayesian methods, which are based on the theory of stochastic processes.

5. Line 62. Substitute "CNN" with "convolutional neural network (CNN)".

6. Line 70. Substitute "DNN" with "deep neural network (DNN)".

7. Line 80. Methods based on the minimization of an objective function, can be improved from the point of view of the computational effort, through the use of the adjoint equation for the computation of the gradient of the objective function. This should be considered by the authors and possibly mentioned or discussed in the manuscript.

8. Lines 83 & 360. Is "designs" the best word? May be, "is based on", "considers" or "proposes"? Similarly for "designed" at line 360.

9. Line 85. Substitute "was" with "is", because the present tense is used in the following sentences.

10. Line 91. Expression "parameter values transition smoothly across space" could be rephrased, possibly as "the spatial variation of parameter values is quite smooth".

11. Line 95. Is "curse" the best word?

12. Lines 95 to 102. These sentences could be improved to explain why different methods have been used for the different scenarios and to motivate the specific choice of each method. This should help to improve the description of what is novel in this work, otherwise the comment by one of the reviewers remains crucial ("The manuscript presents a thorough comparison, but it fails to identify which is the clear innovation brought forward.")

13. Line 136 to 138. These sentences could be rephrased, possibly as "These four methods were proposed at different stages of the development of machine learning, but the application for constructing surrogate models in most groundwater modeling scenarios is still relevant." Did I interpret correctly your thoughts? If so, this sentence remain rather nevertheless rather apodictic and I wonder whether it can be supported in a better way from physical arguments or is it necessary.

14. Line 140. Sentence "The surrogate model for inversion will be constructed using the most accurate among them" remains vague.

15. Line 141. Expression "the values for different simulation components" is not fully clear to me. All the data sets used for the training are normalized with the formula $X_i = \frac{x_i - x_{min}}{x_{max} - x_{min}}$, where $x_i$ is the i-th value of the data set, $x_{min}$ and $x_{max}$ are respectively the minimum and maximum value of the data set, and $X_i$ is the normalized value. Is this right?

16. Equation (4). how is this equation related to the parameters of equations (1) to (3)? Are $x$ and $y$ scalar or vector quantities?

17. Line 150. Substitute "Eq.(5)~(6)", possibly with "equations (5) and (6)".

18. Lines 151 to 155. The notation has to be modified. What is $w^j$? In the second line of equation (6) it could be better to use $(u - \varepsilon)^2$. Remark "$\varepsilon$... insensitive tube" can be erased.

19. Lines 164 & 165. Erase "the penalty parameter" and "the kernel function parameter", the name of the variable is sufficient. However, $\sigma$ is not defined, is it?

20. Equation (7). Do $W$ and $B$ have the same meaning as the same quantities in (4)? $\sigma$ was defined to be a parameter at line 165, here is a function: this is confusing for the Readers who are not familiar with the applied methods. Erase × from the formula.

21. Lines 177 to 204. The notation is unclear, it does not correspond with the notation introduced in the previous part of the manuscript. For instance, symbols $F$ and $G$ have already been used for different quantities. $H$ is not defined is it? The loss function has the same symbol as an hyperparameter of MSVR. $\omega_i$ in equation (12) is not defined, is it?

22. Lines 205 to 215. Is the information about the number of neurons in each hidden layer relevant here, namely, in the description of the methodology? It should be stated later and the motivation for the choice of this value should be given. The same comment applies for the type of activation functions. The whole paragraph could be moved to another point, i.e., after the description of the data sets and where the method is applied.

23. Lines 217 & 218. I partially disagree with statement "the purpose of a surrogate model is to minimize the difference between the predicted outputs and the

numerical modeling outputs": the purpose of a surrogate model is to substitute a high-dimensional model with a low-dimensional model. So the surrogate model must

24. Line 217. Why an L1 norm? L2 norms have been used so far in the work!

25. Line 221. Statement "a widely used machine learning framework" can be erased.

26. Line 226. Symbol $G$ has already been used to denote other quantities, functions, etc.

27. Line 238 & 239. Sentence "For example,... the reduced-dimensional parameters" can be erased, the citation could be sufficient. However, I wonder whether it is the optimal one.

28. Section 2.2.2. Once again the notation is confusing: symbols that have been used previously for some quantities are used here to denote different quantities. Formula $z \sim q(z)$ is given without an explanation.

29. Section 2.3.1. Once again the notation is confusing and sometimes not rigorous. These parts could be moved to the appendix, or, even better in the supplementary material.

30. Section 2.3.2. This section requires a thorough revision, with a clear definition of individual quantities.

31. Line 362 & 366. The measurement unit is a different concept from relevant temporal scales.

32. Line 366. Could "plain" be substituted with "alluvial"?

33. Line 374. Which observation data have been simulated? Hydraulic head? Solute concentrations? This is stated much later only.

34. Lines 374 to 379. The added noise is proportional to the value of the "measured" value. Therefore, this means that the error on hydraulic head is assumed to be very small close to the boundaries where the prescribed head is 0 m and to be the highest at the opposite border of the domain, where the prescribed head attains high values. Unfortunately, hydraulic head represent a potential and as such it could be changed by adding a constant value, without changing the hydraulic gradient, which is the "engine" of groundwater flow. Therefore, if one used a different reference height for hydraulic head, the absolute value of errors on hydraulic head and the errors on hydraulic gradients would change a lot.

35. Line 376. Once again, $\varepsilon$ is used to denote a different quantity.

36. Lines 383, 515, 649. Symbol "~" should be substituted with "to".

37. Sections 3.1 to 3.3. Which method is used for the simulation of flow and transport? Finite differences, finite elements, finite volumes,...? Eulerian or Lagrangian methods for solute transport? Which time spacing is used? Is the transport model purely convective?

38. Lines 390, 402, 423. Words "meshes" or "grids" should be substituted, possibly with "cells" or "elements".
39. Figure 2, Lines 565ff. Here upper case K is used for permeability, whereas lower case k is used in the text. I prefer the latter choice, but a uniform symbol should be used throughout the whole manuscript.
40. Line 394. Word "uncertain" can be erased.
41. Line 406. Expression "are as:" should be corrected.
42. Line 409. Word "stable" should be substituted with "stationary" or "steady-state".
43. Line 410. Add a reference for "equifinality". Indeed, in this way an important prior information and regularization is introduced, without proper discussion.
44. Line 428. Expression "$t$=2~24 years" could be substituted as "from 2 years to 24 year".
45. Line 450. Expression "Figure S3~Figure S6" should be substituted, possibly with "Figures S3 to S6 in the supplementary material".
46. Lines 458ff, Figures 5, 6 & 13, Tables 1 & 3. Measurement units for RMSE are missing. How is RMSE computed for all the model outputs? Head and concentration errors cannot be simply summed up, as they bear different measurement units.
47. Figures 5 and 6. Expression "(a) ~(c) are" should be substituted, possibly with "Plots (a) to (c) show".
48. Section 4.2.1. What is the "logarithmic average convergence" represented in Figure 7? Is it the RMSE?
49. Figure 7. Why the initial value is different among different algorithms? The caption does not specify what is the difference between the four plots. It would be important to recall that the TNNA curve is the same for all the plots. Why is the curve of DE so "noisy"? I have not recognized such an irregular behavior in my experience with that algorithm. The TNNA curve is quite smooth, but it shows very small bumps, in particular slightly after 150 iterations. Is there any explanation for that behavior?
50. Line 505. Is the noise additive or multiplicative? It seems to be additive, now. So there is a difference with respect to what has been described at lines 374 to 379. Why?
51. Lines 516ff. Validation should refer to the use of data sets corresponding to different physical situations from those considered during calibration. So this is not a standard and thorough "model validation".
52. Line 572. Numbers in "K4 and K6" should be subscripts.
53. Figures 8 & 9. The captions do not provide full descriptions of the figure content.
54. Lines 584ff. Once again, "deterministic" is used in a context where the Bayesian, stochastic approach is mentioned.

55. Figure 10. The figure caption should be rewritten. Six rows are mentioned, but the figure has 4 rows and 6 columns. No explanation is given for (a) to (d).
56. Figure 11. The figure caption must be completed with the description of what is represented in the four images.
57. Figure 13. Upper case letter should not be used for measurements units: "days", not "Days".
58. Figure 14. The second row of plot (a) shows a "wavy" behavior. Can it be explained?
59. Section 4.2.3. It is not clear if the values of permeability of the two hydrofacies have been estimated or have been fixed. In other words, which are the parameters to be identified in this tests?
60. Line 669. Expression "Figure 15-16" should be substituted with "Figures 15 and 16".
61. Lines 716 & 717. Sentence "three key aspects should be considered to extended for real-world applications" should be rephrased.
62. Lines 724ff. The statement "heterogeneity exhibits ambiguous statistical features" is not clear to me and this makes it unclear also the following remarks.
63. Line 732 & 733. Expression "such designs are also to eliminate" should be rephrased.
64. Line 763 to 774. These sentences discuss potential future developments, which are not based on the results of this work: therefore, they can be erased.
65. Lines 787, 795, 804, 814, 843, 855, 859, 874, 886, 915, 918, 921, 926, 954, 973, 981. The page numbers or the paper numbers of these scientific articles are missing.
66. Line 835. Volume number and page or paper numbers are missing.
67. Lines 839, 848, 913. Several details are missing for these references.
68. Line 847. Details of this reference should be checked.
69. Lines 850, 904. DOI is missing for these references.
70. Line 908. Details of the reference should be corrected.
71. Line 911. "npj Digital Medicine" should be checked.

---

## Author Response (AR2)

Dear Editor and Reviewers:

Thank you very much for your valuable comments and insightful suggestions on our manuscript. We have carefully considered each comment and revised the manuscript accordingly. Our detailed responses to each comment are listed below.

**Responses to the issues raised by Referee #1 are as follows:**
a. The manuscript fails to identify which is the clear innovation brought forward.
**Response:**
    In the revised manuscript, we have reorganized the second-to-last paragraph of the introduction (lines 86-117). Specifically, the innovation of this study is summarized in line 111-117 as follows:
    "*In summary, the primary contributions of this study are as follows:*
    *(1) Proposed a novel inversion framework that integrates the TNNA algorithm with dimensionality reduction techniques, including KLE and generative machine learning methods, thereby extending its applicability to high-dimensional heterogeneous fields characterized by Gaussian and non-Gaussian stochastic processes, respectively.*
    *(2) Conducted a comprehensive comparative analysis between the TNNA algorithm and four conventional metaheuristic algorithms across three case scenarios, highlighting the advantages of machine learning in inverse estimation under different heterogeneous conditions.*"

b. There is no discussion about parameter sensitivity and hyperparameter optimization of the TNNA algorithm.
**Response:**
    Thank you for your comment. Regarding the hyperparameter optimization for the TNNA algorithm, we did not conduct a formal sensitivity analysis, as the training process of the reverse neural network is guided by the constraints of the inverse objective function, which requires only a set of observation data as input. Therefore, in a GPU hardware environment, we are able to quickly determine suitable hyperparameters based on prior research through an empirical trial-and-error approach.
    Recognizing the significant influence of hyperparameters on neural network performance in some certain scenarios, we have emphasized the importance of hyperparameter optimization in future research (see the last sentence of Section 5):
    "*Furthermore, hyperparameters can significantly influence neural network performance in certain scenarios. It is necessary for future research to explore hyperparameter optimization and sensitivity analysis to identify the optimal neural network structures and training strategies, ultimately enhancing model performance across diverse hydrological conditions.*"

c. There is no sufficient detail about the computational advantage of TNNA with respect to the other techniques (other than saying that you have to run less number of times the forward model for TNNA).
**Response:**
    Thank you for raising this issue. We acknowledge that methodological details supporting these advantages were insufficiently clarified in our original manuscript. In the revised manuscript, we have emphasized the computational characteristics of the four metaheuristic algorithms in the last paragraph of Section 2.3.1, and clarified in Section 2.3.2 why the TNNA algorithm requires only one forward simulation per epoch when training the reverse network:
    "*A common characteristic of all the methods described above is that each iterative update of model parameters requires multiple evaluations of the objective function, and sufficient iterations are necessary to balance local exploitation and global exploration.*" (see the last paragraph of Section 2.3.1)
    ………………
    ………………
    "*In the above process, each backpropagation step involves only a single forward calculation of the loss function. After establishing the computational graph, gradients of the trainable parameters $\theta_{Reverse}$ are computed through backpropagation combined with automatic*

*differentiation. These gradients are then used to update the trainable parameters $\theta_{Reverse}$. Thus, only one forward simulation is executed during each epoch of the reverse network $F_{Reverse}$ training procedure. This presents a marked computational advantage of TNNA compared to the four selected metaheuristic algorithms, which require numerous forward simulations for parameter updates at each iteration.*" (see Section 2.3.2)

The differences in implementation between these two categories of methods, combined with their comparative results presented, clearly illustrate the computational advantage of the TNNA algorithm.

d. There is insufficient detail about how the surrogate models are trained and on which parameters they are trained on.

**Response:**

We appreciate this comment. In Section 4.1, we have provided additional details on the specific data structures of the model parameters and outputs used in the surrogate models for the three case scenarios.

"*Surrogate models were first compared using the Case 1 with low-dimensional parameter. For this scenario, the input parameters for the surrogate models consist of a 9-dimensional vector, including 8 permeability parameters and the contaminant source release concentration. The output consists of the simulated hydraulic heads and solute concentrations at 25 observation points.*" (see the first paragraph in Section 4.1)

"*In the two high-dimensional scenarios, the input parameters for the surrogate models are single-channel matrix data representing the heterogeneous parameter field, while the output consists of vector formed by flattening the multi-channel matrix data, representing the simulated hydraulic heads and solute concentrations at predefined time steps within the simulation domain. The training and testing datasets for these two case scenarios consist of 2000 and 500 samples, respectively.*" (see the last paragraph in Section 4.1, line 503-506)

e. The manuscript is too long, and it has too many details on the different methodologies that could be moved to an appendix to make the reader more comfortable while reading the main part of the text.

**Response:**

Thank you for your suggestion. We have moved the detailed implementation procedures of the metaheuristic algorithms to the supplementary materials and added a summarized paragraph about the four methods in the revised manuscript.

**Responses to the issues raised in the Editor's attachment are as follows:**

1. Lines 13 & 14. Rephrase the sentences, possibly as follows: "Tandem neural network architecture (TNNA) is a machine learning algorithm which has been recently proposed for estimating uncertain parameters with inverse mappings".

**Response:**

Suggestion followed.

2. Lines 22 & 25. Here a percentage of noise is mentioned (1% or 10%), but it is not clearly stated which are the measured quantities and what is used as reference value.

**Response:**

This sentence has been revised as follows:

"*Additionally, we evaluate algorithm performance under two different noise level conditions (multiplicative Gaussian noise with standard deviations of 1% and 10%) for normalized hydraulic head and solute concentration data in the non-Gaussian random field scenario, which exhibits the most complex parameter characteristics.*"

3. Lines 31 to 36. Recent publications only have been considered. However, these concepts are well-established since a long time and can be considered text-book material. Moreover, I wonder whether

the papers referenced for inverse modeling are the most relevant ones. Many other review papers on inverse problems in hydrology are available and should be considered (e.g., 10.1016/0022-1694(87)90207-1, 10.2136/sssabookser5.4.c40, 10.1016/S0167-5648(04)80146-1, 10.2166/nh.2007.024, 10.1029/96WR00160, 10.1007/BF01547729, 10.1016/0309-1708(91)90039-Q, 10.3390/hydrology11110189, 10.1029/WR022i002p00095, 10.1002/hyp.3360060305, and many others).

**Response:**
Suggestion followed.

4. Line 39. I would not use "deterministic" to characterize Bayesian methods, which are based on the theory of stochastic processes.

**Response:**
This sentence has been revised as:
"*Among available algorithms, methods based on objective functions established from maximum a posteriori estimation and solved by optimization techniques represent a significant category*"

5. Line 62. Substitute "CNN" with "convolutional neural network (CNN)".

**Response:**
Suggestion followed.

6. Line 70. Substitute "DNN" with "deep neural network (DNN)".

**Response:**
Suggestion followed.

7. Line 80. Methods based on the minimization of an objective function can be improved, from the point of view of the computational effort, through the use of the adjoint equation for the computation of the gradient of the objective function. This should be considered by the authors and possibly mentioned or discussed in the manuscript.

**Response:**
We appreciate this suggestion and have added the following statements at the end of the surrogate modeling section (lines 65–70):
"*Specifically, inversion approaches based on objective function minimization can also benefit from adjoint methods (Plessix, 2006). Integrating adjoint methods with machine learning-based surrogate models enables efficient gradient computation in high-dimensional and complex scenarios, making their practical implementation tractable (Xiao et al., 2021).*"

8. Lines 83 & 360. Is "designs" the best word? May be, "is based on", "considers" or "proposes"? Similarly for "designed" at line 360.

**Response:**
Thanks for this comment and this word is replaced by "considers".

9. Line 85. Substitute "was" with "is", because the present tense is used in the following sentences.

**Response:**
Suggestion followed.

10. Line 91. Expression "parameter values transition smoothly across space" could be rephrased, possibly as "the spatial variation of parameter values is quite smooth".

**Response:**
We have revised the expression as suggested.

11. Line 95. Is "curse" the best word?

**Response:**
The phrase "curse of dimensionality" is a widely used term in the field of machine learning. However, to avoid potential misunderstanding for readers unfamiliar with this domain, we have rephrased the sentence as follows (line 100):

*"Additionally, dimensionality reduction techniques are necessary for the two high-dimensional cases to reduce computational complexity associated with high-dimensional parameter spaces."*

12. Lines 95 to 102. These sentences could be improved to explain why different methods have been used for the different scenarios and to motivate the specific choice of each method. This should help to improve the description of what is novel in this work, otherwise the comment by one of the reviewers remains crucial ("The manuscript presents a thorough comparison, but it fails to identify which is the clear innovation brought forward.")

**Response:**

Thank you for this suggestion. We have added the reason for choosing KLE and generative machine learning methods for dimensionality reduction for Gaussian random fields and non-Gaussian random fields, respectively:

*"Specifically, the Karhunen-Loève Expansion (KLE) method is feasible for Gaussian random fields. It reconstructs the Gaussian random field through a linear combination of orthogonal basis functions, …… These methods can establish relationships between low-dimensional standard distributions (e.g., uniform distribution) and high-dimensional distributions, effectively representing non-Gaussian random fields as low-dimensional latent vectors (i.e., parameters after dimensionality reduction)."*

For the innovation of this study, we have revised the description as follows:

*"In summary, the primary contributions of this study are as follows:*

*(1) Proposed a novel inversion framework that integrates the TNNA algorithm with dimensionality reduction techniques, including KLE and generative machine learning methods, thereby extending its applicability to high-dimensional heterogeneous fields characterized by Gaussian and non-Gaussian stochastic processes, respectively.*

*(2) Conducted a comprehensive comparative analysis between the TNNA algorithm and four conventional metaheuristic algorithms across three case scenarios, highlighting the advantages of machine learning in inverse estimation under different heterogeneous conditions."*

13. Line 136 to 138. These sentences could be rephrased, possibly as "These four methods were proposed at different stages of the development of machine learning, but the application for constructing surrogate models in most groundwater modeling scenarios is still relevant." Did I interpret correctly your thoughts? If so, this sentence remain rather nevertheless rather apodictic and I wonder whether it can be supported in a better way from physical arguments or is it necessary.

**Response:**

Thank you for this suggestion. We agree that the original statement was somewhat general and could be better clarified. To address this, we have revised the description to directly introduce the four machine learning models along with their respective architectural characteristics. The revised text reads as follows:

*"Specifically, four popular machine learning models with distinct architectural differences are evaluated for surrogate modeling. These are: multi-output support vector regression (MSVR), a kernel-based architecture for data mapping; fully connected deep neural network (FC-DNN), composed of stacked fully connected layers; LeNet, a classical convolutional neural network (CNN) architecture; and deep residual convolutional neural network (ResNet), which incorporates residual connections into the CNN structure."*

14. Line 140. Sentence "The surrogate model for inversion will be constructed using the most accurate among them" remains vague.

**Response:**

This sentence has been rephrased as: *"The predictive accuracy of four surrogate modeling approaches will be compared in this study, and the best-performing approach among them will subsequently be selected for inversion computations."* (line 158)

15. Line 141. Expression "the values for different simulation components" is not fully clear to me. All the data sets used for the training are normalized with the formula $X_i =(x_i−x_{min})/(x_{max}−x_{min})$, where $x_i$ is the i-th value of the data set, $x_{min}$ and $x_{max}$ are respectively the minimum and maximum

value of the data set, and $X_i$ is the normalized value. Is this right?

**Response:**

Yes, your understanding is correct. We have revised the original sentence as follows and added a reference describing the specific normalization method:

"*Before constructing surrogate models, the training datasets are normalized separately for each simulation component using Min-Max Normalization, in which each component is scaled independently based on its minimum and maximum values, ensuring that all normalized values fall within the range [0, 1] (Chen et al., 2021).*" (line 159)

16. Equation (4). how is this equation related to the parameters of equations (1) to (3)? Are $x$ and $y$ scalar or vector quantities?

**Response:**

Thank you for this helpful comment. We have revised the notation in Equation (4): the original $x$ and $y$ are replaced with $\boldsymbol{m}$ and $\hat{\mathbf{y}}$ to maintain consistency with Equations (1) to (3).

17. Line 150. Substitute "Eq.(5)~(6)", possibly with "equations (5) and (6)".

**Response:**

Suggestion followed.

18. Lines 151 to 155. The notation has to be modified. What is $w^j$ ? In the second line of equation (6) it could be better to use $(u - \varepsilon)^2$. Remark "$\varepsilon$… insensitive tube" can be erased.

**Response:**

Thank you for this suggestion. The notation $w^j$ represents the regression vector within matrix $W$ corresponding to the $j^{\text{th}}$ observed dataset. We have revised the definitions of $W$ and $B$ in the manuscript as follows: $W=[w^1,\ldots,w^{N_{obs}}]^T \in \mathbb{R}^{N_{obs} \times N_{samples}}$ and $B=[b^1,\ldots,b^{N_{obs}}]^T \in \mathbb{R}^{N_{obs} \times 1}$. Equation (6) has been updated to use the term $(u - \varepsilon)^2$, and the remark "$\varepsilon$… insensitive tube" has been removed accordingly.

19. Lines 164 & 165. Erase "the penalty parameter" and "the kernel function parameter", the name of the variable is sufficient. However, $\sigma$ is not defined, is it?

**Response:**

Thank you for this comment. We have erased the phrases "the penalty parameter" and "the kernel function parameter" as suggested. In addition, $\sigma$ is a bandwidth parameter of the kernel function. So, we supplemented the definition of the kernel function explicitly in the revised manuscript.

"*where $F_{MSVR}(\boldsymbol{m})$ denotes the dataset regression model operator constructed based on MSVR; $\varphi(\boldsymbol{m})$ is a nonlinear regression function that implicitly maps the input vector $\boldsymbol{m}$ into a high-dimensional feature space. Its inner product defines the kernel function $K(\boldsymbol{m}, \boldsymbol{m}_i)$ (here we use the Gaussian radial basis function (RBF) kernel: $K(\boldsymbol{m}, \boldsymbol{m}_i) = \varphi(\boldsymbol{m})^T\varphi(\boldsymbol{m}_i) = exp(-0.5 \| \boldsymbol{m} - \boldsymbol{m}_i \|^2 / \sigma^2))$.*" (line 171)

20. Equation (7). Do $W$ and $B$ have the same meaning as the same quantities in (4)? $\sigma$ was defined to be a parameter at line 165, here is a function: this is confusing for the Readers who are not familiar with the applied methods. Erase × from the formula.

**Response:**

We have revised Equation (7) by changing the notations $W$ and $B$ to $W_{DNN}$ and $B_{DNN}$, respectively, to explicitly indicate that they represent the weight parameter matrix and bias vector of the fully-connected layer in DNNs. Additionally, we have updated the notation of the $l$th activation function to $f_{\sigma-l}(\cdot)$.

21. Lines 177 to 204. The notation is unclear, it does not correspond with the notation introduced in the previous part of the manuscript. For instance, symbols $F$ and $G$ have already been used for different quantities. $H$ is not defined is it? The loss function has the same symbol as an hyperparameter of MSVR. $\omega i$ in equation (12) is not defined, is it?

**Response:**

Thank you for your suggestion. We have revised the notation accordingly in the revised

manuscript.

22. Lines 205 to 215. Is the information about the number of neurons in each hidden layer relevant here, namely, in the description of the methodology? It should be stated later and the motivation for the choice of this value should be given. The same comment applies for the type of activation functions. The whole paragraph could be moved to another point, i.e., after the description of the data sets and where the method is applied.
**Response:**
We appreciate your valuable comment. The determination of the optimal number of hidden layers for the FC-DNN has been moved to Section 4.1. The description of the two CNNs (LeNet and ResNet) architectures remain in the methodology section, as these architectures are fixed throughout this study. The description regarding the hidden-layer design of the FC-DNN has been revised as follows:
"*The performance of the FC-DNN is sensitive to the number of hidden layers, whose optimal value is determined based on specific case studies presented in the application section.*"

The motivations behind the selection of activation functions are supplemented as follows:
"*The activation function for the output layer is Sigmoid to constrain outputs within the range of 0 to 1. For hidden layers, the Swish activation function is adopted due to its smooth form with non-monotonic and continuously differentiable properties, which helps improve the DNN training procedures (Elfwing et al., 2018).*"

Additionally, the explanation of the hidden-layer selection in Section 4.1 has been updated accordingly:
"*For the FC-DNN, the optimal number of hidden layers was separately determined for each of the four datasets. The candidate range for the number was set from 1 to 7. According to the $RMSE_{All}$ and $R^2_{All}$ values in Table S2 and Table S3, optimal number of hidden layers for in the FC-DNN for $D_{train-200}$, $D_{train-500}$, $D_{train-1000}$ and $D_{train-2000}$ are 2, 4, 3, and 3, respectively.*"

23. Lines 217 & 218. I partially disagree with statement "the purpose of a surrogate model is to minimize the difference between the predicted outputs and the numerical modeling outputs": the purpose of a surrogate model is to substitute a high-dimensional model with a low-dimensional model. So the surrogate model must provide outputs which closely resemble those of a high-dimensional model.
**Response:**
Thank you for highlighting this issue. This statement may lead to misunderstandings. Our intention here was to introduce the formulation of the loss function for the surrogate model. Therefore, we have revised the original text as follows:
"*The surrogate models are trained by minimizing the difference between the predicted outputs $\hat{y}_i = F_{DNN}(\boldsymbol{m}_i, \theta_{DNN})$ and the numerical modeling outputs $y_i$. Following prior researches (Mo et al., 2019, 2020; Chen et al., 2021), the loss function is formulated with L1 norm constraints:*" (line 252-254)

24. Line 217. Why an L1 norm? L2 norms have been used so far in the work!
**Response:**
We are thankful for this comment. The difference between L1 norm and L2 norm is that L1 norm is based on Laplace distribution hypothesis and L2 norm is based on Gaussian distribution hypothesis. For the observation noises are considered as Gaussian distribution in this paper, the objective function for inversion is based on L2 norm. While training surrogate models with L1-norm is primarily based on recommendations from previous studies and insights gained from our own research experience. In fact, L2-norm is also widely used and may be applicable for this study. In the revised manuscript, the relevant reference citations are added.
"*Following prior researches (Mo et al., 2019, 2020; Chen et al., 2021), the loss function is formulated with L1 norm constraints:*" (line 252-254)

25. Line 221. Statement "a widely used machine learning framework" can be erased.
**Response:**

Suggestion followed.

26. Line 226. Symbol $G$ has already been used to denote other quantities, functions, etc.
**Response:**
The symbol $G(s)$ is replaced by $Y_G(s)$.

27. Line 238 & 239. Sentence "For example,… the reduced-dimensional parameters" can be erased, the citation could be sufficient. However, I wonder whether it is the optimal one.
**Response:**
This sentence has been removed, and we have added references Loève (1955) and Mariethoz and Caers (2014) for the Karhunen–Loève expansion. (line 273)

28. Section 2.2.2. Once again the notation is confusing: symbols that have been used previously for some quantities are used here to denote different quantities. Formula $z \sim q(z)$ is given without an explanation.
**Response:**
We have revised the symbols used in the manuscript, unifying the representation of the low-dimensional vectors in Sections 2.2.1 and 2.2.2 as $z$. The explanation for the $z \sim q(z)$ is supplemented as:
*"The distribution of the latent vectors $\{z_1,…, z_N\}$, obtained by mapping the N prior model parameter samples $\{m_1,…, m_N\}$, is denoted as $q(z)$."* (line 278-279)

29. Section 2.3.1. Once again the notation is confusing and sometimes not rigorous. These parts could be moved to the appendix, or, even better in the supplementary material.
**Response:**
We have revised the notation in this section, and have moved the detailed steps of metaheuristic algorithms to the supplementary material. A summarized paragraph is retained in the main text as follows:
*"The four metaheuristic algorithms used in this paper essentially update model parameters through distinct heuristic stochastic search strategies. …………Simulated Annealing (SA) starts from a random initial solution and iteratively explores neighbouring solutions, accepting them probabilistically based on the Metropolis criterion, while gradually decreasing temperature parameter until convergence (Metropolis et al., 1953; Kirkpatrick et al., 1983).*
*A common characteristic of all the methods described above is that each iterative update of model parameters requires multiple evaluations of the objective function, and sufficient iterations are necessary to balance local exploitation and global exploration. Detailed implementation procedures and theoretical foundations of these methods are provided in the supplementary materials. The metaheuristic algorithms used in this study were implemented using the open-source Python package scikit-opt (https://scikit-opt.github.io/ )."*

30. Section 2.3.2. This section requires a thorough revision, with a clear definition of individual quantities.
**Response:**
Suggestion followed. The definitions of individual quantities in Section 2.3.2 have been revised.

31. Line 362 & 366. "Measurement unit" is a different concept from"relevant temporal scale".
**Response:**
To avoid potential confusion with the term "scale," we have revised the manuscript to explicitly describe the differences in geometric sizes among the cases.
*"Both Case 1 and Case 2 are approximately tens of meters in size, with simulation time measured in days.…………Case 3 simulates contaminant plume migration, has a size of approximately one kilometre, and simulation time measured in years."* (line 357-358)

32. Line 366. Could "plain" be substituted with "alluvial"?
**Response:**
"Plain" has been replaced with "alluvial" as it is indeed more appropriate.

33. Line 374. Which observation data have been simulated? Hydraulic head? Solute concentrations? This is stated much later only.

**Response:**

Thank you for the comment. We have clarified this in the manuscript as follows (line 369):

*"The observation data (hydraulic heads and solute concentrations) for model parameter inversion are generated by adding Gaussian noise perturbations to the numerical model simulation results."*

34. Lines 374 to 379. The added noise is proportional to the value of the "measured" value. Therefore, this means that the error on hydraulic head is assumed to be very small close to the boundaries where the prescribed head is 0 m and to be the highest at the opposite border of the domain, where the prescribed head attains high values. Unfortunately, hydraulic head represent a potential and as such it could be changed by adding a constant value, without changing the hydraulic gradient, which is the "engine" of groundwater flow. Therefore, if one used a different reference height for hydraulic head, the absolute value of errors on hydraulic head and the errors on hydraulic gradients would change a lot.

**Response:**

Thanks for this valuable comment. I fully agree with your point of view, and your insights will provide significant inspiration for our future research. We admitted that our original description led to your misunderstanding. The observation noise in this study was added after data normalization. Thus, no matter how large the measured hydraulic head values are, their normalized values always range from 0 to 1, and these normalized values directly reflect the relative differences in hydraulic head. Additionally, the primarily purpose of this study is to examine how varying noise levels affect the inversion results. Applying multiplicative noise with different standard deviations provides a feasible method to design two distinct observational noise levels. To avoid misunderstandings, we have explicitly clarified in the revised manuscript that noise was introduced based on the normalized numerical simulation results:

"*Specifically, observational noise is introduced by multiplying the **min-max normalized** simulated data by a random noise factor $\epsilon \sim N(1, \sigma^2)$,*" (line 370)

35. Line 376. Once again, $\varepsilon$ is used to denote a different quantity.

**Response:**

The "$\varepsilon$" is replaced by "$\epsilon$" here.

36. Lines 383, 515, 649. Symbol "~" should be substituted with "to".

**Response:**

Suggestion followed.

37. Sections 3.1 to 3.3. Which method is used for the simulation of flow and transport? Finite differences, finite elements, finite volumes,…? Eulerian or Lagrangian methods for solute transport? Which time spacing is used? Is the transport model purely convective?

**Response:**

Thank you for this comment. We have supplemented additional information about the forward modeling solution in the last sentence of the first paragraph of Section 3, as follows:

*"The numerical models of the three cases are established using TOUGHREACT, which employs an integral finite difference method with sequential iteration procedures and adaptive time stepping to solve the flow and transport equations. Dispersion effects are inherently incorporated through molecular diffusion and numerical dispersion induced by upstream weighting and grid discretization (Xu et al., 2011)."* (line 362-365)

38. Lines 390, 402, 423. Words "meshes" or "grids" should be substituted, possibly with "cells" or "elements".

**Response:**

Thank you for this comment. We have uniformly replaced "meshes" and "grids" with "cells" throughout the manuscript.

39. Figure 2, Lines 565ff. Here upper case K is used for permeability, whereas lower case k is used in the text. I prefer the latter choice, but a uniform symbol should be used throughout the whole manuscript.

**Response:**

We have uniformly revised the notation to use the lowercase *k* throughout the manuscript.

40. Line 394. Word "uncertain" can be erased.

**Response:**

Suggestion followed.

41. Line 406. Expression "are as:" should be corrected.

**Response:**

"are as:" has been modified as "are"

42. Line 409. Word "stable" should be substituted with "stationary" or "steady-state".

**Response:**

The word "stable" has been replaced with "stationary" as recommended.

43. Line 410. Add a reference for "equifinality". Indeed, in this way an important prior information and regularization is introduced, without proper discussion.

**Response:**

Suggestion followed. We have added relevant discussions and references:

"*It should be noted that in high-dimensional parameter scenarios, the increased degrees of freedom typically result in greater parameter uncertainty. Insufficient observational information may fail to effectively constrain parameter estimation, resulting in potential uncertainty and equifinality (Mclaughlin and Townley, 1996; Zhang et al., 2015; Cao et al., 2025). ...........introducing these constraints ensures the stability and robustness of the inversion outcomes without affecting the inherent performance characteristics of the five optimization algorithms compared in this study.*" (line 406-413)

44. Line 428. Expression "t=2~24 years" could be substituted as "from 2 years to 24 year".

**Response:**

Suggestion followed.

45. Line 450. Expression "Figure S3~Figure S6" should be substituted, possibly with "Figures S3 to S6 in the supplementary material".

**Response:**

Suggestion followed.

46. Lines 458ff, Figures 5, 6 & 13, Tables 1 & 3. Measurement units for RMSE are missing. How is RMSE computed for all the model outputs? Head and concentration errors cannot be simply summed up, as they bear different measurement units.

**Response:**

When calculating RMSE, both hydraulic head and solute concentration data are normalized to a unified scale between 0 and 1. Consequently, the RMSE values are dimensionless and have no specific measurement units. To clarify this point, we added the following statement at the end of the second paragraph in Section 4.1:

"*Additionally, it should be noted that the above RMSE and $R^2$ metrics are computed based on the normalized hydraulic head and solute concentration data.*" (line 454-455)

47. Figures 5 and 6. Expression "(a) ~(c) are" should be substituted, possibly with "Plots (a) to (c) show".

**Response:**

Suggestion followed.

48. Section 4.2.1. What is the "logarithmic average convergence" represented in Figure 7? Is it the RMSE?

**Response:**

"logarithmic average convergence" represents logarithmic objective function values during inversion iterations. We have added a clarification in line 513:

"*Figure 7 presents the logarithmic average convergence curves (i.e., log10 of the average objective value during inversion iterations) of four metaheuristic algorithms and the TNNA algorithm throughout 100 parameter scenarios.*" (line 523-524)

49. Figure 7. Why the initial value is different among different algorithms? The caption does not specify what is the difference between the four plots. It would be important to recall that the TNNA curve is the same for all the plots. Why is the curve of DE so "noisy"? I have not recognized such an irregular behavior in my experience with that algorithm. The TNNA curve is quite smooth, but it shows very small bumps, in particular slightly after 150 iterations. Is there any explanation for that behavior?

**Response:**

Thank you for your insightful comments and questions. Below, we provide clarifications regarding your concerns:

The differences in initial values arise from the distinct initialization strategies of the algorithms. For the four metaheuristic algorithms (DE, GA, PSO, SA), the initial objective value corresponds to the best among $N_{PC}$ candidates randomly sampled from the prior distribution. In this study, each algorithm was run independently without a fixed seed, resulting in slight variations due to randomness, though the values remain within a similar range. In contrast, the initial model parameters of TNNA method are not directly sampled, but are determined by the randomly initialized weights of the reverse network. Therefore, its initial objective value typically differs significantly from those of the metaheuristic algorithms. For the purpose of this study, the inconsistency in initial points does not affect the comparison of results.

The caption for Figure 7 has been revised as:

"***Figure 7.*** *Comparative convergence trends (log$_{10}$ of the average objective value) of five optimization algorithms on 100 parameter scenarios. Plot (a)~(d) compare the four metaheuristic algorithms and TNNA under $N_{PC}$=20, 40, 60, and 80, respectively; TNNA was executed only once on the same 100 parameter scenarios, and its curve is identical across all plots; Markers indicate convergence values every 10 iterations.*"

Regarding the issue of the noisy curve of DE, model parameter optimization by metaheuristic algorithms is a stochastic process, and it is normal for fluctuations in objective function values to occur during convergence processes. For the DE method, when $N_{PC}$=80 for instance, the objective function values after 150 iterations range between $9.05 \times 10^{-5}$~$1.32 \times 10^{-4}$ (corresponding to logarithmic values of -4.04~-3.88 in Figure 7(d)). Fluctuations between consecutive iterations typically remain within $1 \times 10^{-5}$ (mostly around $3 \times 10^{-6}$), which is a reasonable magnitude for optimization algorithms. The DE curve appears more noticeably noisy in Figure 7 due to its relatively larger fluctuation amplitude compared to other methods.

50. Line 505. Is the noise additive or multiplicative? It seems to be additive, now. So there is a difference with respect to what has been described at lines 374 to 379. Why?

**Response:**

Thank you very much for pointing out this issue. We used multiplicative noise in all numerical examples presented in this study. We have also rechecked our computational procedures and confirmed that this inconsistency was a typo. We have corrected it in the revised manuscript accordingly.

51. Lines 516ff. Validation should refer to the use of data sets corresponding to different physical situations from those considered during calibration. So this is not a standard and thorough "model validation".

**Response:**

Thank you very much for this clarification. We acknowledge that our understanding and usage of the term "validation" were indeed not precise enough. To avoid any misunderstanding, we have revised the description related to this aspect accordingly in the manuscript.

52. Line 572. Numbers in "K4 and K6" should be subscripts.
**Response:**
Suggestion followed.

53. Figures 8 & 9. The captions do not provide full descriptions of the figure content.
**Response:**
Suggestion followed.

54. Lines 584ff. Once again, "deterministic" is used in a context where the Bayesian, stochastic approach is mentioned.
**Response:**
Thank you for your comment. The "deterministic" is replaced by "well-defined".

55. Figure 10. The figure caption should be rewritten. Six rows are mentioned, but the figure has 4 rows and 6 columns. No explanation is given for (a) to (d).
**Response:**
The figure caption has been revised as:
    *"Spatial distributions of log-permeability field estimation results (row 1, 3, and 5 for $N_{PC}$=100, 500, and 1000, respectively) and absolute errors (row 2, 4, and 6 for $N_{PC}$=100, 500, and 1000, respectively) for Scenario 5, achieved by four metaheuristic algorithms (plots (a) to (d) correspond to GA, DE, PSO and SA, respectively)."*

56. Figure 11. The figure caption must be completed with the description of what is represented in the four images.
**Response:**
    The figure caption has been revised as:
    *"Figure 11. Spatial distributions log-permeability field estimation results and absolute errors for Scenario 5, achieved by the TNNA. Plots (a) and (c) show the log-permeability fields estimated using 1000 (TNNA-1000) and 200 (TNNA-200) training samples, respectively; plots (b) and (d) present the corresponding absolute error distributions."*

57. Figure 13. Upper case letter should not be used for measurements units: "days", not"Days".
**Response:**
    Suggestion followed.

58. Figure 14. The second row of plot (a) shows a "wavy" behavior. Can it be explained?
**Response:**
    This "wavy" behavior primarily results from the numerical precision of the simulated hydraulic head data. In this study, the hydraulic head simulation precision is 0.01 m (i.e., 1 cm), which means that the minimum scale of simulated error is also 0.01 m. Given that the color bar for displaying hydraulic head errors ranges from 0 to 0.1m, this discretization at intervals of 0.01 m creates a visual "wavy" pattern.

59. Section 4.2.3. It is not clear if the values of permeability of the two hydrofacies have been estimated or have been fixed. In other words, which are the parameters to be identified in this tests?
**Response:**
    We appreciate this comment and apologize for the lack of clarity. In this case, the permeability values of the two hydrofacies were fixed. The parameters to be identified are the hydrofacies types assigned to each discrete grid cell, essentially formulates a high-dimensional binary inverse problem. We have supplemented this clarification at the end of the first paragraph of Section 3.3 as follows:
    *"This case focus on a high-dimensional binary inverse problem aimed at identifying the lithofacies type of each discrete grid cell within the domain. Note that the permeability values of the two lithofacies are fixed in this case."* (line 429-431)

60. Line 669. Expression "Figure 15-16" should be substituted with "Figures 15 and 16".
**Response:**

Suggestion followed.

61. Lines 716 & 717. Sentence "three key aspects should be considered to extended for real-world applications" should be rephrased.
**Response:**
Thank you for this suggestion. The sentence has been revised to:
"*Given the complexities of subsurface systems, three key aspects should be considered to extend the TNNA method to real-world applications.*" (line 736-737)

62. Lines 724ff. The statement "heterogeneity exhibits ambiguous statistical features" is not clear to me and this makes it unclear also the following remarks.
**Response:**
This part aims to emphasize that the primary challenge faced in practical research is obtaining representative parameter field samples. The original description has now been revised accordingly:
"*Generative machine learning methods (including state-of-the-art variants) also have the potential to characterize more complex non-Gaussian fields. However, obtaining representative parameter field datasets remains challenging in practical research. For instance, spatial variations in non-stationary stochastic aquifer systems may result in significant discrepancies in geostatistical parameters across sampling windows (Mariethoz and Caers, 2014). Therefore, developing appropriate generator training strategies is essential for these practical scenarios.*" (line 742-747)

63. Line 732 & 733. Expression "such designs are also to eliminate" should be rephrased.
**Response:**
Thank you for this comment. This sentence has been revised to
"*Such monitoring strategies for comparing inversion methods also aim to minimize external interferences, ensuring that performance differences are primarily determined by inversion algorithms themselves.*" (line 751-753)

64. Line 763 to 774. These sentences discuss potential future developments, which are not based on the results of this work: therefore, they can be erased.
**Response:**
Suggestion followed.

65. Lines 787, 795, 804, 814, 843, 855, 859, 874, 886, 915, 918, 921, 926, 954, 973, 981. The page numbers or the paper numbers of these scientific articles are missing.
**Response:**
This missing page numbers have been completed. Note that the references originally listed in lines 804 and 843 have been deleted along with the corresponding paragraphs, as suggested in comment 64.

66. Line 835. Volume number and page or paper numbers are missing.
**Response:**
This reference has been deleted along with the corresponding paragraphs, as suggested in comment 64.

67. Lines 839, 848, 913. Several details are missing for these references.
**Response:**
Missed details have been supplemented, primarily including the conference locations and DOI information.

68. Line 847. Details of this reference should be checked.
**Response:**
Done.

69. Lines 850, 904. DOI is missing for these references.
**Response:**
The DOI for these references have been supplemented.

70. Line 908. Details of the reference should be corrected.
**Response:**
   Suggestion followed.

71. Line 911. "npj Digital Medicine" should be checked.
**Response:**
   This reference has been deleted along with the corresponding paragraphs, as suggested in comment 64.

---

## Editor Decision (ED2)

The authors followed the reviewers' suggestions and answered most of my comments.

Overall, I appreciate their effort, but some comments have not been addressed in a fully satisfactory way and some residual language or typo mistakes should be fixed before accepting the manuscript for publication.

1. The answer to comment "b. There is no discussion about parameter sensitivity and hyperparameter optimization of the TNNA algorithm" by referee #1 is not satisfactory. I warmly ask the authors to carefully reconsider this issue.

2. The answer to comment "d. There is insufficient detail about how the surrogate models are trained and on which parameters they are trained on" by Referee #1 is not fully satisfactory. Some information has been added, but I am afraid that the information is not sufficient for a reader who would like to apply the same procedure.

3. Lines 370ff. the description of the noise added to the data is now very clear. Unfortunately, my comment remains still valid and basically not answered. In fact, if $x$ is the "true" physical quantity, then the normalized value is
$X = (x - x_{min})(x_{max} - x_{min})^{-1}$ and $x = x_{min} + X(x_{max} - x_{min})$.
The noisy normalized value is $X' = (1 + \delta)X$, where $\delta$ follows a normal distribution, with zero mean and unit standard deviation. The corresponding noisy physical quantity is given by $x' = x_{min} + (1 + \delta)X(x_{max} - x_{min})$.
The absolute error on the physical quantity is then $x' - x = \delta X(x_{max} - x_{min})$. It is clear from this formula, that the absolute error is proportional to $X$, so that it is negligible for values of $x$ close to $x_{min}$. On the other hand it is maximum, when $x$ is close to $x_{max}$. I do not see any physical reason for such a choice.
Moreover, the relative error is given by
$$\frac{x' - x}{x} = \frac{\delta X(x_{max} - x_{min})}{x_{min} + X(x_{max} - x_{min})}$$
If $x_{min} = 0$, then the relative error is equal to $\delta$, otherwise it depends also on the value of $X$.
I think that a proper discussion of this issue is very important.

4. The answer to the comment on Figure 14, namely on the "wavy" behavior shown in the second row of plot (a), opens a relevant question. If I understood properly, the hydraulic head has been simulated with a threshold of 0.01 m on the iterative method applied to solve the flow equation: if this is the case, the simulation error is greater than 0.01 m. However, let us assume that this is the order of magnitude of the simulation error on "noise-free" heads. Than, this is the same order of magnitude as the error added in the low-noise tests, when a 1% standard deviation is considered. In other words, the noise-free data share an error with the same order of magnitude

as the noise in the low-level tests. However, the basic difference is that the simulation error could have the same absolute error everywhere, whereas the added noise is proportional to $X$, as demonstrated above. Am I wrong?

5. Line 43 & 44. Sentence "Among available algorithms, methods based on objective functions established from maximum a posteriori estimation and solved by optimization techniques represent a significant category" remains quite ambiguous. May be, it could be substituted with "Methods based on the minimization of objective functions or the maximization of posterior distributions require the application of optimization techniques".

6. Lines 67 to 70. Indeed, my comment intended to stress that the use of the adjoint equation limits the number of runs of the forward problem for the application of gradient-based optimization algorithms.

7. Line 115. The sentence should be rephrased. It might be ambiguous to what word "respectively" refers.

8. Line 117. Specify which is the benchmark with respect to which ML methods give an advantage.

9. Lines 119 & 120. Sentence "With advancements... for future studies" could be erased.

10. Line 141. $z$ is not defined, is it?

11. Line 145, equation (3). If I understand correctly the notation, the operator for parameter dimensionality reduction $G$ computes the high-dimensional parameter vector $m$ starting from a low-dimensional vector $z$. Then the computed vector $m$ is used in the forward (high-fidelity?) model. I think this is not the proper description of surrogate models.

12. Line 150. Expression "calculating their responses" should be rephrased.

13. Line 155. Expression "convolutional neural network" can be erased, because the acronym CNN has already been defined.

14. Line 162. Is the reference Chen et al. (2021) significant? The min-max normalization was introduced long time before that paper.

15. Lines 173ff. $m_i$ is not defined, is it?

16. Line 237. Other choices, e.g., hyperbolic tangent, could provide output in the range from 0 to 1. So, why the Sigmoid was chosen?

17. Lines 253 & 254. The motivation for the use of L1 norm for the loss function is not very informative. Moreover, is "constraints" the right word here?

18. Line 286. $G$ has already been used at line 141, even if here it is a scalar and there it was a vector.

19. Line 333. "After obtain" should be corrected as "After obtaining".

20. Line 338. Word "vector" should be corrected as "vector".

21. Line 355. Expression "measured in days" should be substituted, possibly with "of 60 days".
22. Line 358. Expression "measured in year" should be substitute, possibly with "of several years (up to 30 years)".
23. Line 409. The concept of equifinality has been introduced in hydrology at least in 1992 by Beven & Binley (DOI: 10.1002/hyp.3360060305).
24. Line 429. Word "focus" should be corrected as "focuses".
25. Line 524. Word "during" should be substituted, possibly with "as a function of the".
26. Lines 555ff, Figure 7. The remark about the "noisy" curve of DE that the authors included in their "Response to comments" should be added to the text of the manuscript.
27. Lines 740ff. Punctuation should be checked.
28. Line 743. Words "also" and "more" could be erased.
29. Lines 743 & 744. Expression "representative parameter field datasets" should be clarified.
30. Line 761. Meteorological factors can affect head measurements, but other factors could be much more relevant.

---

## Author Response (AR3)

Dear Editor:

Thank you very much for your valuable comments and insightful suggestions on our manuscript. We have carefully considered each comment and revised the manuscript accordingly. Our detailed responses to each comment are listed below.

1. The answer to comment "b. There is no discussion about parameter sensitivity and hyperparameter optimization of the TNNA algorithm" by referee #1 is not satisfactory. I warmly ask the authors to carefully reconsider this issue.
**Response:**
    Thanks for this comment. We have added a paragraph at the end of Section 2.1.2 clarifying the critical hyperparameters affecting DNN training when using the Adam optimizer within a PyTorch implementation. Specifically, the paragraph discusses how these hyperparameters influence the training outcomes and highlights that their optimal values for different scenarios were determined through a trial-and-error approach. The added paragraph reads as follow:
    "*When conducting DNN training, hyperparameter selection primarily influences the update process of trainable parameters. Besides the weight decay mentioned above, learning rate and the number of epochs are two other crucial hyperparameters directly affecting training stability and convergence speed. A larger learning rate accelerates initial convergence but may lead to oscillations near the optimal solution, whereas a smaller learning rate tends to improve final accuracy but requires more epochs to achieve convergence. In this study, we first set a relatively large number of epochs to ensure that the trainable parameters are adequately updated. Subsequently, appropriate learning rates and weight decay values for different scenarios are determined through a trail-and-error approach.*"

2. The answer to comment "d. There is insufficient detail about how the surrogate models are trained and on which parameters they are trained on" by Referee #1 is not fully satisfactory. Some information has been added, but I am afraid that the information is not sufficient for a reader who would like to apply the same procedure.
**Response:**
    Thanks for this comment. We have added the hyperparameter settings used for training the neural network models as follows:
    "*When training the FC-DNN, LeNet, and ResNet for Case 1, the hyperparameters for batch size and learning rate were consistently set to 50 and $1 \times 10^{-4}$, respectively. The weight decay values for LeNet and ResNet were both set to $1 \times 10^{-5}$, while FC-DNN used a weight decay of 0. The number of training epochs was uniformly set to 500 for all three models.*"(Line 507-509)

    "*For ResNet training in Case 2 (Gaussian random field), the hyperparameters were set as follows: batch size =100, learning rate =$1 \times 10^{-4}$, and weight decay =$1 \times 10^{-6}$. For Case 3 (non-Gaussian random field), the corresponding values were batch size = 50, learning rate = $1 \times 10^{-3}$, and weight decay= $1 \times 10^{-8}$. In both cases, the number of training epochs was also set to 500.*" (Line 555-557)

3. Lines 370ff. the description of the noise added to the data is now very clear. Unfortunately, my comment remains still valid and basically not answered. In fact, if is the "true" physical quantity, then the normalized value is $X = (x - x_{min})(x_{max} - x_{min})^{-1}$ and $x = x_{min} + X(x_{max} - x_{min})$. The noisy normalized value is $X' = (1 + \delta)X$, where $\delta$ follows a normal distribution, with zero mean and unit standard deviation. The corresponding noisy physical quantity is given by $x' = x_{min} + (1+\delta)X(x_{max} - x_{min})$.
The absolute error on the physical quantity is then $x' - x = \delta X(x_{max} - x_{min})$. It is clear from this formula, that the absolute error is proportional to $X$, so that it is negligible for values of $x$ close to $xmin$. On the other hand it is maximum, when $x$ is close to $x_{max}$. I do not see any physical reason for such a choice. Moreover, the relative error is given by

$$\frac{x' - x}{x} = \frac{\delta X(x_{max} - x_{min})}{x_{min} + X(x_{max} - x_{min})}$$

If $x_{min} = 0$, then the relative error is equal to $\delta$, otherwise it depends also on the value of $X$.
I think that a proper discussion of this issue is very important.

**Response:**

We appreciate this detailed and insightful comment regarding the formulation of the multiplicative noise model and its implications for the distribution of errors in the physical domain. We agree that this type of noise leads to absolute errors that are proportional to the normalized values, resulting in negligible errors near the lower bound ($x_{\min}$) and more significant errors near the upper bound ($x_{\max}$).

The multiplicative noise model adopted in this study was based on its usage in some previous studies, and the physical motivation behind this choice was not thoroughly examined at the time. Based on the literature reviewed during this revision, we recognize that the multiplicative noise model has the practical advantage of ensuring non-negativity of perturbed observations, which is important near plume boundaries with low concentrations. Additionally, whether observation noise depends on the measured values in practice is often determined by the specific measurement technique employed. In the revised manuscript, we have supplemented the discussion of this issue, including two measurement scenarios where multiplicative noise may naturally arise in real-world environmental applications. The complete revised paragraph added to the manuscript is as follows:

*"Here we applied the multiplicative noise is intended to ensure that all perturbed observation values remain non-negative, which is particularly important in regions near plume boundaries where concentrations are close to zero. Generally, observation errors are assumed to be independent of the measured values, whereas the multiplicative noise model introduces value-proportional perturbations, resulting in a positive correlation between the standard deviation of observation noise and the true values. This type of error dependence may also exist in real-world studies when certain measurement techniques are used. For example, in hydraulic head monitoring, pressure transducers may exhibit drift (i.e., a persistent deviation in output not caused by actual pressure changes) due to aging and fatigue of components such as the diaphragm or strain gauge, leading to reduced measurement accuracy (Sorensen and Butcher, 2011). Variation in hydraulic pressure can lead to different levels of drift among transducers, with those installed at higher pressure (i.e., high hydraulic heads) environments tending to experience more significant drift and thus being more prone to elevated observation noise. For the analysis of solute concentrations in laboratory settings, when the concentrations of water samples exceed the detection range of the instrument, a common approach is to dilute these samples prior to measurement. While analytical instruments may introduce additive errors at a relatively fixed level, the rescaling process following dilution (i.e., multiplying the measured value by the dilution factor) amplifies these errors. As a result, the final measurement error becomes approximately proportional to the original solute concentration (Kabala and Skaggs, 1998). Given that the goal of this study is to evaluate the robustness of five inversion algorithms under different noise levels, both additive and multiplicative noise models are suitable for representing observational uncertainty. Prior work by Neupauer et al. (2000) demonstrated that the choice between these two noise types has minimal influence on the comparative performance of inversion methods."*

4. The answer to the comment on Figure 14, namely on the "wavy" behavior shown in the second row of plot (a), opens a relevant question. If I understood properly, the hydraulic head has been simulated with a threshold of 0.01 m on the iterative method applied to solve the flow equation: if this is the case, the simulation error is greater than 0.01 m. However, let us assume that this is the order of magnitude of the simulation error on "noise-free" heads. Then, this is the same order of magnitude as the error added in the low-noise tests, when a 1% standard deviation is considered. In other words, the noise-free data share an error with the same order of magnitude as the noise in the low-level tests. However, the basic difference is that the simulation error could have the same absolute error everywhere, whereas the added noise is proportional to $X$, as demonstrated above. Am I wrong?

**Response:**

We appreciate this insightful comment. The numerical accuracy in this study is controlled based on the relative error in the conservation equations. Specifically, we set the convergence criterion to a relative error threshold of $10^{-5}$, meaning that the maximum local mass imbalance in any grid cell does not exceed one part in 100,000 of the total mass in that cell. After completing the numerical simulation that meets this convergence criterion, the hydraulic head outputs were

recorded with a precision of one centimeter. We have added a clarification regarding the solver's numerical precision in lines 380–385 of the revised manuscript, as follows:

"*In all the three cases, the relative error tolerance for the conservation equations was uniformly set to $10^{-5}$, ensuring that the maximum imbalance of conserved quantities within each discrete grid cell remains below one part in 100,000 of the total quantity in that cell.*"

The results compared in Figure 4 represent the absolute differences between numerical model outputs generated using the true parameter field and those obtained using the parameter fields estimated through inversion. These error distributions do not reflect numerical approximation errors, nor do they include the added multiplicative noise. In Lines 700–702, we have explicitly clarified the meaning of symbols used in Figure 4 as follows:

"*Note that in Figure 14, $\hat{y}_H$ and $\hat{y}_C$ represent the simulated spatial distributions of hydraulic heads and solute concentrations based on the estimated permeability fields through inverse modeling, while $y_H$ and $y_C$ represent those simulated using the true permeability field.*"

5. Line 43 & 44. Sentence "Among available algorithms, methods based on objective functions established from maximum a posteriori estimation and solved by optimization techniques represent a significant category" remains quite ambiguous. May be, it could be substituted with "Methods based on the minimization of objective functions or the maximization of posterior distributions require the application of optimization techniques".

**Response:**

Suggestion followed.

6. Lines 67 to 70. Indeed, my comment intended to stress that the use of the adjoint equation limits the number of runs of the forward problem for the application of gradient-based optimization algorithms.

**Response:**

Thank you for this comment. We have added a discussion at the end of the second paragraph of the Introduction regarding the application of adjoint equations in optimization algorithms, and highlighted potential implementation challenges including the overwhelming programming effort and complexity involved in deriving adjoint equations:

"*The efficiency of optimization algorithms can be enhanced by integrating them with adjoint methods, particularly when extended to high-dimensional parameter spaces. Adjoint methods are capable of efficiently computing gradients for all parameters simultaneously through solving adjoint equations derived from the original forward model (Plessix, 2006). This gradient information can directly accelerate local optimization algorithms (Epp et al., 2023) and facilitate gradient-enhanced global optimization methods (Kapsoulis et al., 2018), significantly improving efficiency in complex inverse problems. However, practical implementation of adjoint methods remains challenging due to the overwhelming programming effort and the complexity associated with deriving adjoint equations, especially for strong nonlinear system models (Xiao et al., 2021; Ghelichkhan et al., 2024).*"

In the third paragraph of the Introduction, the advantages of integrating adjoint methods with DNN-based surrogate models was revised as follows:

"*Additionally, due to their inherent differentiability and continuity, DNN-based surrogate models can be integrated with adjoint equations, enabling efficient gradient computations, and significantly facilitating their practical implementation in high-dimensional and complex scenarios (Xiao et al., 2021).*"

7. Line 115. The sentence should be rephrased. It might be ambiguous to what word "respectively" refers.

**Response:**

Thank you for this comment. This sentence has been revised as follows:

"*Proposed a novel inversion framework that integrates the TNNA algorithm with dimensionality reduction techniques, including KLE for Gaussian stochastic processes and*

*generative machine learning methods for non-Gaussian stochastic processes, thereby extending its applicability to high-dimensional heterogeneous fields."*

8. Line 117. Specify which is the benchmark with respect to which ML methods give an advantage.
**Response:**
    We appreciate this comment. The benchmark here refers to "metaheuristic stochastic search strategies", and the machine learning methods mentioned here correspond to "DNN-based reverse mapping". Accordingly, this sentence is revised as:
    *"Conducted a comprehensive comparative analysis between the TNNA algorithm and four conventional metaheuristic algorithms across three case scenarios, highlighting the advantages of DNN-based reverse mapping over metaheuristic stochastic search strategies for inverse estimation under different heterogeneous conditions."*

9. Lines 119 & 120. Sentence "With advancements… for future studies" could be erased.
**Response:**
    Suggestion followed.

10. Line 141. $z$ is not defined, is it?
**Response:**
    Thank you for this comment. The definition of $z$ has been supplemented as follows:
    *"…………, where $z$ is a low-dimensional vector whose parameter space is commonly defined as an easily sampled probability distribution (e.g., standard Gaussian or uniform distribution)."*

11. Line 145, equation (3). If I understand correctly the notation, the operator for parameter dimensionality reduction $G$ computes the high-dimensional parameter vector $m$ starting from a low-dimensional vector $z$. Then the computed vector $m$ is used in the forward (high-fidelity?) model. I think this is not the proper description of surrogate models.
**Response:**
    Thank you for this comment. We understand your concern and recognize that our original wording might have caused confusion. To clarify, in high-dimensional parameter scenarios, it is necessary not only to construct a surrogate model to enhance the computational efficiency of forward simulations, but also to address the dimensionality reduction of high-dimensional model parameters. We have enriched the original paragraph with more details, explicitly explaining how the dimensionality-reduced parameter representation allows indirect inversion based on Equation (3). The revised paragraph is as follows:
    *"In high-dimensional parameter scenarios, directly optimizing the model parameter $m$ can lead to computational difficulties due to its high dimensionality. To mitigate this issue, in addition to constructing a surrogate model $F_{Forward}(\cdot)$ to improve the computational efficiency of forward simulations,………".*

12. Line 150. Expression "calculating their responses" should be rephrased.
**Response:**
    Thanks for this comment. This sentence is revised as follows.
    *"The process begins by sampling model parameters from prior distributions. The corresponding system responses for these parameter samples are simulated using a high-fidelity numerical model."*

13. Line 155. Expression "convolutional neural network" can be erased, because the acronym CNN has already been defined.
**Response:**
    Suggestion followed.

14. Line 162. Is the reference Chen et al. (2021) significant? The min-max normalization was introduced long time before that paper.

**Response:**

Thanks for this comment. We agree that min-max normalization (0–1 normalization) has indeed been introduced much earlier than Chen et al. (2021). The purpose of citing this paper here was primarily to provide a recent reference that clearly describes the complete formulation and implementation details, particularly regarding data normalization methods specifically tailored for groundwater model parameter inversion involving multiple simulated components. This is beneficial for readers to directly understand the detailed implementation processes. We have accordingly clarified this in the revised manuscript by adding the following note:

"*Details regarding the specific formulation and implementation of normalization in groundwater models involving multiple simulated components can be found in Chen et al. (2021)*"

15. Lines 173ff. $\boldsymbol{m}_i$ is not defined, is it?

**Response:**

Thank you for this comment. The definition of $\boldsymbol{m}_i$ has been supplemented as follows:

"*……$\boldsymbol{m}_i$ denotes the ith model parameter vector from the surrogate model training dataset).*"

16. Line 237. Other choices, e.g., hyperbolic tangent, could provide output in the range from 0 to 1. So, why the Sigmoid was chosen?

**Response:**

Thank you for this insightful comment. Indeed, the primary consideration in choosing the activation function for the output layer is ensuring that the output values align with the target value range. We agree that other activation functions, such as the hyperbolic tangent (-1 to 1) and ReLU (0 to $+\infty$), could also theoretically serve as suitable output-layer activations, and these functions have also proven effective in practice to guarantee surrogate model accuracy. We specifically chose the Sigmoid function to strictly constrain initial model outputs within the target range (0–1), thereby reducing the risk of occasional extreme or anomalous predictions, particularly in the early stages of training. We have included the following explanation in the revised manuscript:

"*Note that other activation functions whose outputs cover this range can also be adopted, ……, thereby reducing the risk of occasional extreme or anomalous predictions, particularly in the early stages of training*"

17. Lines 253 & 254. The motivation for the use of L1 norm for the loss function is not very informative. Moreover, is "constraints" the right word here?

**Response:**

We appreciate this comment. This sentence related to the word of "constraints" has been modified as:

"*……, the L1 norm-based loss function is adopted and formulated as:*"

We have added the following description to clearly illustrate the implications of selecting the L1 or L2 norm for surrogate modeling:

"*It should be note that the L2 norm can also be employed as a loss function in constructing surrogate model tasks. Due to its squared-error formulation, the L2 norm provides smoother gradients and more stable parameter updates near convergence compared to the L1 norm; however, this formulation also makes it more sensitive to extreme outliers. When the sampled parameters sparsely cover the parameter space, adopting the L1 norm loss function can improve the robustness of surrogate model predictions.*"

18. Line 286. $G$ has already been used at line 141, even if here it is a scalar and there it was a vector.

**Response:**

The "$G$" has been replaced by "$\mathcal{G}$"

19. Line 333. "After obtain" should be corrected as "After obtaining".
**Response:**
Thanks for this comment. We have corrected it.

20. Line 338. Word "vetcor" should be corrected as "vector".
**Response:**
Done.

21. Line 355. Expression "measured in days" should be substituted, possibly with "of 60 days".
**Response:**
Suggestion followed.

22. Line 358. Expression "measured in year" should be substitute, possibly with "of several years (up to 30 years)".
**Response:**
Suggestion followed.

23. Line 409. The concept of equifinality has been introduced in hydrology at least in 1992 by Beven & Binley (DOI: 10.1002/hyp.3360060305).
**Response:**
This reference has been added accordingly.

24. Line 429. Word "focus" should be corrected as "focuses".
**Response:**
Thanks for this comment. The "focus" has been corrected as "focuses".

25. Line 524. Word "during" should be substituted, possibly with "as a function of the".
**Response:**
Suggestion followed.

26. Lines 555ff, Figure 7. The remark about the "noisy" curve of DE that the authors included in their "Response to comments" should be added to the text of the manuscript.
**Response:**
Suggestion followed. The added sentences are as:
*"It is worth noting that the five optimization algorithms rely on stochastic processes for parameter updates. Therefore, the objective function values are not guaranteed to decrease monotonically with each iteration. According to Figure 7, the DE algorithm exhibits more noticeable fluctuations compared to other algorithms. Nevertheless, these fluctuations remain within a reasonable range. For example, at $N_{PC}=80$, the objective function values after 150 iterations range between $9.05 \times 10^{-5} \sim 1.32 \times 10^{-4}$ (corresponding to logarithmic values of $-4.04 \sim -3.88$ in Figure 7(d)). Fluctuations between consecutive iterations typically remain within $1 \times 10^{-5}$ (mostly around $3 \times 10^{-6}$), which is considered reasonable for optimization algorithms."*

27. Lines 740ff. Punctuation should be checked.
**Response:**
Done.

28. Line 743. Words "also" and "more" could be erased.
**Response:**
Suggestion followed.

29. Lines 743 & 744. Expression "representative parameter field datasets" should be clarified.
**Response:**
    This sentence is revised as follows:
    "*However, obtaining representative parameter field datasets that accurately capture the spatial variability and heterogeneous geostatistical characteristics of the target aquifer remains challenging in practical research.*"

30. Line 761. Meteorological factors can affect head measurements, but other factors could be much more relevant.
**Response:**
    We appreciate this comment acknowledge that the original description was overly absolute. This sentence was revised as follows:
    "*Similarly, hydraulic head measurements may be influenced by other factors, including meteorological conditions, human groundwater extraction, and engineering disturbances, among others.*"

---

## Editor Decision (ED3)

Line 59. I would erase "the overwhelming programming effort and", because in many cases, the adjoint equation has the same form as the equations to be solved for the forward problem, so that the same code can be used for both the forward problem and the adjoint equation.

Lines 172 & 173. Sentence "Details… in Chen et al. (2021).." can be erased.

Line 271. Correct "note" as "noted".

Line 286. Correct "trail-and-error".

Lines 408 & 409. Expression "Here we applied the multiplicative noise is intended" should be corrected.

Lines 413 to 418. The effect of a possible instrumental drift is not strictly related to the proportionality of the error on the measured value.

Line 610. "range between $9.05 \times 10^{-5} \sim 1.32 \times 10^{-4}$ (corresponding to logarithmic values of -4.04~-3.88" should be substituted with "range between $9.05 \times 10^{-5}$ and $1.32 \times 10^{-4}$ (corresponding to logarithmic values between -4.04 and -3.88".

---

## Author Response (AR4)

1. Line 59. I would erase "the overwhelming programming effort and", because in many cases, the adjoint equation has the same form as the equations to be solved for the forward problem, so that the same code can be used for both the forward problem and the adjoint equation.
**Response:**
Suggestion followed.

2. Lines 172 & 173. Sentence "Details… in Chen et al. (2021).." can be erased.
**Response:**
This sentence has been erased.

3. Line 271. Correct "note" as "noted".
**Response:**
Done.

4. Line 286. Correct "trail-and-error".
**Response:**
Thank you for pointing this out. The phrase has been corrected to "*trial-and-error*".

5. Lines 408 & 409. Expression "Here we applied the multiplicative noise is intended" should be corrected.
**Response:**
Thank you for this comment. This sentence has been corrected as "*Here, we applied the multiplicative noise to……*"

6. Lines 413 to 418. The effect of a possible instrumental drift is not strictly related to the proportionality of the error on the measured value.
**Response:**
Thank you for this comment. To prevent potential misunderstanding, we have modified the expression "*tending to experience more significant drift and thus being more prone to*" to "*tending to experience more significant drift, which in some cases may result in elevated observation noise*", in order to clarify that this is a potential rather than a strictly deterministic relationship.

7. Line 610. "range between 9.05×10-5~1.32×10-4 (corresponding to logarithmic values of -4.04~-3.88" should be substituted with "range between 9.05×10-5 and 1.32×10-4 (corresponding to logarithmic values between -4.04 and -3.88".
**Response:**
Done.